# CERTIFIABLY ROBUST INTERPRETATION IN DEEP LEARNING

## ABSTRACT

Deep learning interpretation is essential to explain the reasoning behind model predictions. Understanding the robustness of interpretation methods is important especially in sensitive domains such as medical applications since interpretation results are often used in downstream tasks. Although gradient-based saliency maps are popular methods for deep learning interpretation, recent works show that they can be vulnerable to adversarial attacks. In this paper, we address this problem and provide a certifiable defense method for deep learning interpretation. We show that a *sparsified* version of the popular *SmoothGrad* method, which computes the average saliency maps over random perturbations of the input, is certifiably robust against adversarial perturbations. We obtain this result by extending recent bounds for certifiably robust smooth classifiers to the interpretation setting. Experiments on ImageNet samples validate our theory.

## 1 INTRODUCTION

### 1.1 MOTIVATION

The growing use of deep learning in a wide range of highly-sensitive applications, such as autonomous driving, medicine, finance, and even the legal system (Teichmann et al., 2018; Quellec et al., 2017; Fischer & Krauss, 2018; Nissan, 2017), raises concerns about human trust in machine learning systems. Interpretations, which explain why certain predictions are made, are critical for establishing this trust between users and the machine learning system. Moreover, the interpretation results themselves are often used for downstream tasks. For example, gradient-based saliency maps have been used in medical applications: Quellec et al. (2017) uses gradient-based interpretation techniques to highlight signs of retinal disease in retinal fundus photographs (see Figure 1-b.) Similarly, BenTaieb et al. (2016) uses a gradient-based saliency map as part of a pipeline to automatically analyse histopathology slides of colon tumours. Techniques like these are also used in medical research to annotate datasets (Lahiani et al., 2019). Outside of the medical field, gradient-based interpretations can be used in general-purpose image segmentation tasks (Hong et al., 2015).

However, Ghorbani et al. (2019) has shown that several gradient-based interpretation methods are sensitive to adversarial examples, obtained by adding a small perturbation to the input image. These adversarial examples maintain the original class label while greatly distorting the saliency map (Figure 1-a: see 'Gradient' interpretation method). Although adversarial attacks and defenses on image classification have been studied extensively in recent years (Szegedy et al., 2014; Uesato et al., 2018; Goodfellow et al., 2015; Athalye et al., 2018a;b; Buckman et al., 2018; Kurakin et al., 2017; Papernot & McDaniel, 2016; Carlini & Wagner, 2017; Madry et al., 2018), less attention has been paid to effectively defending deep learning interpretation against adversarial examples (Lee et al., 2019; Etmann et al., 2019). This is partially due to the difficulty of protecting high-dimensional saliency maps compared to defending a class label, as well as to the lack of a ground truth for interpretation. Interestingly, we have observed that the standard adversarial training used for the *classification* robustness does *not* lead to robust interpretations (see Section 4). This calls for better understanding the robustness of interpretation methods and developing defenses with performance guarantees.

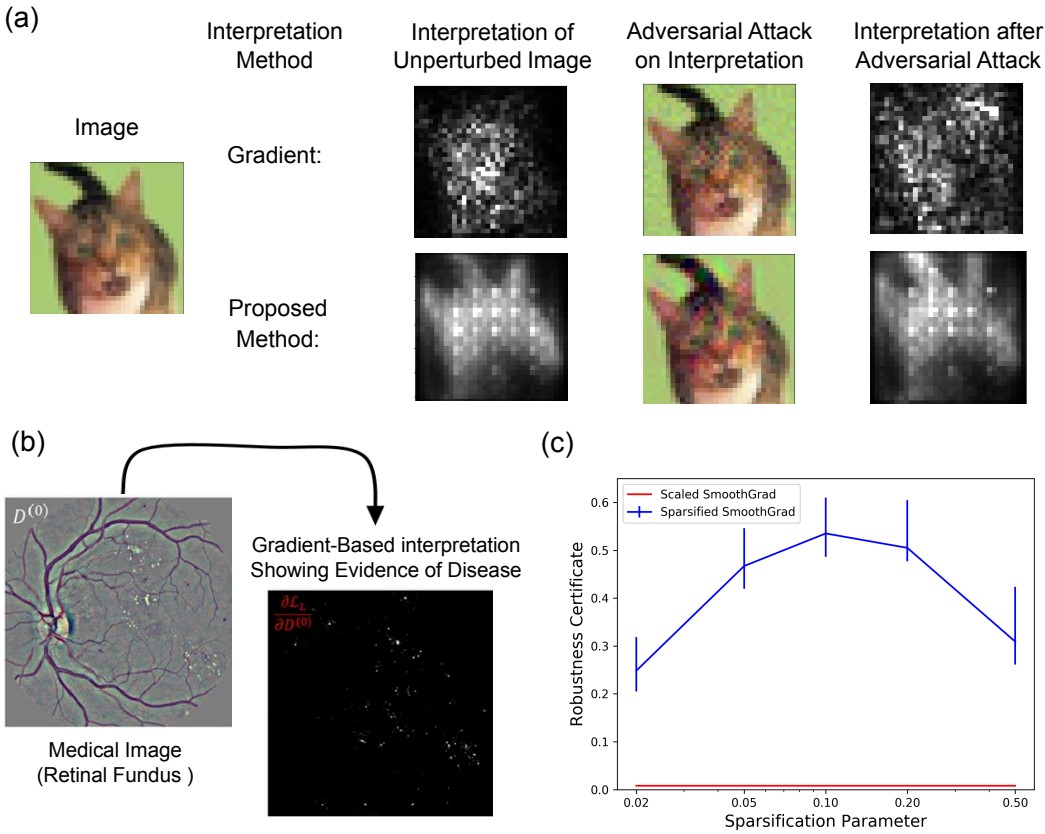

Figure 1: (a) An illustration of the sensitivity of gradient-based saliency maps to an adversarial perturbation of an image from CIFAR-10. Our proposed method, Sparsified SmoothGrad (relaxed variant shown here; see Section 2.3), produces saliency maps which considerably are more robust to adversarial attack than plain gradient saliency maps. (b) A medical use of gradient-based saliency maps. Figures are borrowed from Quellec et al. (2017). Signs of lesions indicative of diabetic retinopathy are automatically highlighted. (c) A comparison of robustness certificate values $R^{\mathrm{cert}}/K$ of Sparsified SmoothGrad vs. scaled SmoothGrad, on ImageNet samples.

## 1.2 PROPOSED APPROACH AND MAIN CONTRIBUTIONS

In the last couple of years, several approaches have been proposed for interpreting neural network outputs (Simonyan et al., 2014; Sundararajan et al., 2017; Alvarez-Melis & Jaakkola, 2018; Adebayo et al., 2018; Huang et al., 2017). Specifically, Simonyan et al. (2014) computes the elementwise absolute value of the gradient of the largest class score with respect to the input. To define some notation, let $\mathbf{g}(\mathbf{x})$ be this most basic form of the gradient-based saliency map, for an input image $\mathbf{x} \in \mathbb{R}^n$. For simplicity, we also assume that elements of $\mathbf{g}(\mathbf{x})$ have been linearly normalized to be between 0 and 1. $\mathbf{g}(\mathbf{x})$ represents, to a first order linear approximation, the importance of each pixel in determining the class label. Numerous variations of this method have been introduced, which we review in the appendix.

A popular saliency map method which extends the basic gradient method is SmoothGrad (Smilkov et al., 2017), which takes the average gradient over random perturbations of the input. Formally, we define the smoothing function as:

$$\bar{\mathbf{g}}(\mathbf{x}) \coloneqq \mathbb{E}\left[\mathbf{g}(\mathbf{x} + \epsilon)\right], \tag{1.1}$$

where $\epsilon$ has a normal distribution (i.e. $\epsilon \sim \mathcal{N}(0, \sigma^2 \mathbf{I})$). We will discuss other smoothing functions in Section 2.3 while the empirical smoothing function which computes the average over finitely many

perturbations of the input will be discussed in Section 2.2. We refer to the basic method described in the above equation as the *scaled* SmoothGrad [1].

Since a ground truth label for interpretation is not available, we use a similarity metric between the original and perturbed saliency maps as an estimate of the interpretation robustness. Ghorbani et al. (2019) introduced a *top-K overlap* metric for this purpose: define $R(\mathbf{x}, \tilde{\mathbf{x}}, K)$ as the number of overlapping elements between top $K$ largest elements of saliency maps of $\mathbf{x}$ and the perturbed image $\tilde{\mathbf{x}}$. Ghorbani et al. (2019) developed an $L_\infty$ attack on $R(\mathbf{x}, \tilde{\mathbf{x}}, K)$, which we adapt to the $L_2$ case to test empirical robustness: see the appendix for details. This top-K overlap metric is naturally motivated: absolute magnitude has little meaning in saliency maps, which are nearly always presented as normalized. Moreover, it is common (Sundararajan et al., 2017) to clip the highest values when displaying saliency maps, so that more than a few pixels are clearly visible. This suggests that relative ranks of salience values (and not their absolute values) are important for interpretation.

Note that for an input $\mathbf{x}$, $R(\mathbf{x}, \tilde{\mathbf{x}}, K)$ depends on the specific perturbation $\tilde{\mathbf{x}}$. We define $R^*(\mathbf{x}, K)$ as the robustness measure with respect to the *worst* perturbation of $\mathbf{x}$. That is,

$$R^*(\mathbf{x}, K) := \min_{\tilde{\mathbf{x}}} \ R(\mathbf{x}, \tilde{\mathbf{x}}, K) \tag{1.2}$$
$$\|\tilde{\mathbf{x}} - \mathbf{x}\|_2 \le \rho.$$

For deep learning models, this optimization is non-convex in general. Thus, characterizing the true robustness of interpretation methods is likely intractable for most modern classifier models.

In this paper, we show that a lower bound on the true robustness value of an interpretation method (i.e. a robustness certificate) can be computed efficiently. In other words, for a given input $\mathbf{x}$, we compute a robustness certificate $R^{\text{cert}}$ such that $R^{\text{cert}}(\mathbf{x}, K) \le R^*(\mathbf{x}, K)$. To establish the robustness certificate for saliency map methods, we prove the following result for a general function $\mathbf{h}(.)$ whose range is between 0 and 1:

**Theorem 2.** *Let $\mathbf{h}(\mathbf{x})$ be the output of an interpretation method whose range is between 0 and 1 and let $\bar{\mathbf{h}}$ be its smoothed version defined as in equation 1.1. Let $\bar{\mathbf{h}}_i(\mathbf{x})$ and $\bar{\mathbf{h}}_{[i]}(\mathbf{x})$ be the $i$-th element and the $i$-th largest elements of $\bar{\mathbf{h}}(\mathbf{x})$, respectively. Let $\Phi$ be the cdf of the normal distribution. If*

$$\Phi\left(\Phi^{-1}\left(\bar{\mathbf{h}}_{[i]}(\mathbf{x})\right) - \frac{2\rho}{\sigma}\right) \ge \bar{\mathbf{h}}_{[2K-i]}(\mathbf{x}), \tag{1.3}$$

*then for the smoothed interpretation method, we have $R^{cert}(\mathbf{x}, K) \ge i$.*

Intuitively, this means that, if there is a sufficiently large *gap* between the $i$-th largest element of the smoothed saliency map and its $(2K - i)$-th largest element, then we can certify that at least $i$ elements in the top $K$ largest elements of the original smoothed saliency map will also be in the top $K$ elements of adversarially perturbed saliency map. We present a more general version of this result with *empirical* expectations for smoothing in Section 2, as well as another rank-based robustness certificate in Section 3. The proof of this bound relies on an extension of Cohen et al. (2019) which addresses certified robustness in the classification case. Proofs for all theorems are given in the appendix.

Evaluating the robustness certificate for the scaled SmoothGrad method on ImageNet samples produces vacuous bounds (Figure 1-c). This motivated us to develop variations of SmoothGrad with larger robustness certificates. One such variation is *Sparsified SmoothGrad* which is defined by smoothing a sparsification function that maps the largest elements of $\mathbf{g}(\mathbf{x})$ to one and the rest to zero. Sparsified SmoothGrad obtains a considerably larger value of the robustness certificate (Figure 1-c) while producing high-quality saliency maps (Figure 3). We study other variations of Sparsified SmoothGrad in Section 2.

A considerably different certification result for robust interpretation is provided by Lee et al. (2019). That work provides a certified $L_2$ radius in which a first-order gradient saliency map is *exactly identical* to the saliency map for the unperturbed image. The method depends on the network structure being locally linear (ReLU-based): the inner radius of the locally linear region is directly calculated.

---

[1]The original definition of SmoothGrad does not normalize and take the absolute values of gradient elements before averaging. We start with the definition of equation 1.1 because it can more easily be compared to our certifiably robust interpretation scheme discussed in Section 2.

By contrast, our method makes no demands on the structure of the classifier, and could be applied to saliency maps generated by methods other than computing simple gradients: any bounded saliency function can be used as $\mathbf{h}(\mathbf{x})$ in our Theorem 1. Additionally, our certification method is computationally inexpensive enough to allow certificates to be computed on ImageNet-scale samples.

## 2 SMOOTHING FOR CERTIFIABLY ROBUST INTERPRETATION

### 2.1 NOTATION

We introduce the following notations to indicate Gaussian smoothing: for a function $\mathbf{h}$, we define population and empirical smoothed functions, respectively, as:

$$
\begin{aligned}
\bar{\mathbf{h}}(\mathbf{x}) &= \mathbb{E}_{\epsilon \sim \mathcal{N}(0,\sigma^2 I)}[h(\mathbf{x}+\epsilon)], \\
\tilde{\mathbf{h}}(\mathbf{x}) &= \frac{1}{q}\sum_{i=1}^{q} h(\mathbf{x}+\epsilon_i), \quad \epsilon_i \sim \mathcal{N}(0,\sigma^2 I).
\end{aligned}
\tag{2.1}
$$

In other words, $\bar{\mathbf{h}}(\mathbf{x})$ represents the expected value of $\mathbf{h}(\mathbf{x})$ when smoothed under normal perturbations of $\epsilon$ with some standard deviation $\sigma$ while $\tilde{\mathbf{h}}(\mathbf{x})$ represents an empirical estimate of $\bar{\mathbf{h}}(\mathbf{x})$ using $q$ samples. We call $\sigma^2$ the smoothing variance and $q$ the number of smoothing perturbations.

We use $\mathbf{v}_i$ to denote the $i^{th}$ element of the vector $\mathbf{v}$. Similarly $\mathbf{h}_i(\mathbf{x})$ denotes the $i^{th}$ element of the output $\mathbf{h}(\mathbf{x})$. We also define, for any $\mathbf{h}(\mathbf{x})$, $\mathrm{rank}(\mathbf{h}(\mathbf{x}),i)$ as the ordinal rank of $\mathbf{h}_i(\mathbf{x})$ in $\mathbf{h}(\mathbf{x})$ (in the descending order): $\mathrm{rank}(\mathbf{h}(\mathbf{x}),i) = j$ denotes that $\mathbf{h}_i(\mathbf{x})$ is the $j^{th}$ largest element in $\mathbf{h}(\mathbf{x})$. We use $\mathbf{x}_{[i]}$ to denote the $i^{th}$ largest element in $\mathbf{x}$. If $i$ is not an integer, the ceiling of $i$ is used.

### 2.2 ROBUSTNESS CERTIFICATE

In order to derive a robustness certificate for saliency maps, we present an extension of the classification robustness result of Cohen et al. (2019) to real-valued functions, rather than discrete classification functions. In our case, we will apply this to the saliency map vector $\mathbf{g}$. First, we define a *floor* function to simplify notation.

**Definition 2.1.** The Floor function is a function $L : [0,1] \rightarrow [0,1]$, such that

$$
L(z) = \Phi\left(\Phi^{-1}(z) - \frac{2\rho}{\sigma}\right)
$$

where $\rho$ denotes the $L_2$ norm of the adversarial distortion and $\sigma^2$ denotes the smoothing variance. $\Phi$ is the cdf function for the standard normal distribution and $\Phi^{-1}$ is its inverse.

Below (Theorem 1) is a general result which can be used to derive robustness certificates for interpretation methods. In particular, it is used to derive our robustness certificate, Theorem 2, later in the section:

**Theorem 1.** *Let $\mathbf{h} : \mathbb{R}^n \rightarrow [0,1]^n$ be a real-valued function. Let $L(.)$ be the floor function defined as in equation 2.1 with parameters $\sigma^2$ and $\rho$. Using $\sigma^2 \in \mathbb{R}$ as the smoothing variance for $\mathbf{h}$, $\forall\, i,j \in [n]$, $\mathbf{x}, \tilde{\mathbf{x}} \in \mathbb{R}^n$ where $\|\mathbf{x}-\tilde{\mathbf{x}}\| \le \rho$:*

$$
L\left(\bar{\mathbf{h}}_i(\mathbf{x})\right) \ge \bar{\mathbf{h}}_j(\mathbf{x}) \Rightarrow \bar{\mathbf{h}}_i(\tilde{\mathbf{x}}) \ge \bar{\mathbf{h}}_j(\tilde{\mathbf{x}}).
$$

Note that this theorem is valid for any general function. However, we will use it for our case where $\bar{\mathbf{h}}(\mathbf{x})$ is a smoothed saliency map. Theorem 1 states that, for a given saliency map vector $\bar{\mathbf{h}}(\mathbf{x})$, if $L(\bar{\mathbf{h}}_i(\mathbf{x})) \ge \bar{\mathbf{h}}_j(\mathbf{x})$, then if $\mathbf{x}$ is perturbed inside an $L_2$ norm ball of radius at most $\rho$, $\bar{\mathbf{h}}_i(\tilde{\mathbf{x}}) \ge \bar{\mathbf{h}}_j(\tilde{\mathbf{x}})$. This result extends Theorem 1 in Cohen et al. (2019) in two ways: first, it provides a guarantee about the difference in the values of two quantities, which in general might not be related, while the original result compared probabilities of two mutually exclusive events. Second, we are considering a real-valued function $\mathbf{h}$, rather than a classification output which can only take discrete values. This bound can be compared directly to Lecuyer et al. (2019)'s result which similarly concerns unrelated elements in a vector. Just as in the classification case (as noted by Cohen et al. (2019)), Theorem 1 gives a significantly tighter bound than that of Lecuyer et al. (2019) (see details in the appendix).

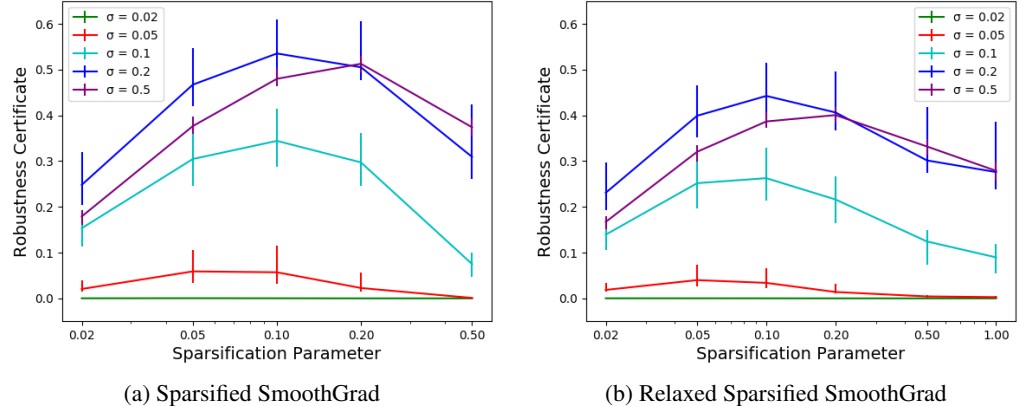

(a) Sparsified SmoothGrad

(b) Relaxed Sparsified SmoothGrad

Figure 2: Certified robustness bounds on ImageNet for different values of the sparsification parameter $\tau$. The lines shown are for the $60^{th}$ percentile guarantee, meaning that 60 percent of images had guarantees at least as tight as those shown. For both examples, $K = 0.2n$, and $\rho = 0.03$ (in units where pixel intensity varies from 0 to 1.)

We can extend Theorem 1 to use empirical estimates of smoothed functions. Following Lecuyer et al. (2019), we derive upper and lower bounds of the expected value function $\bar{\mathbf{h}}(\mathbf{x})$ in terms of $\tilde{\mathbf{h}}(\mathbf{x})$, by applying Hoeffding's Lemma. To present our result for the empirical case, we first define an *empirical floor function* to derive a similar lower bound when the population mean is estimated using a finite number of samples:

**Definition 2.2.** The Empirical Floor function is a function $\hat{L} : [0,1] \to [0,1]$, such that for given values of $\rho$, $\sigma$, $p$, $q$, $n$, where $\rho$ denotes the maximum $L_2$ distortion, $\sigma^2$ denotes the smoothing variance, $p$ denotes the probability bound, $q$ denotes the number of perturbations, and $n$ is the size of input of the function:

$$\hat{L}(z) = \Phi\left(\Phi^{-1}\left(z - c\right) - \frac{2\rho}{\sigma}\right) - c, \quad \text{where } c := \sqrt{\frac{\ln(2n(1-p)^{-1})}{2q}}.$$

**Corollary 1.** *Let* $\mathbf{h} : \mathbb{R}^n \to [0,1]^n$ *be a function such that for given values of* $q$, $\sigma$, $\forall i, j \in [n]$, $\mathbf{x}$, $\tilde{\mathbf{x}} \in \mathbb{R}^n$, $\|\mathbf{x} - \tilde{\mathbf{x}}\|_2 \le \rho$, *with probability at least* $p$,

$$\hat{L}(\tilde{\mathbf{h}}_i(\mathbf{x})) \ge \tilde{\mathbf{h}}_j(\mathbf{x}) \Rightarrow \bar{\mathbf{h}}_i(\tilde{\mathbf{x}}) \ge \bar{\mathbf{h}}_j(\tilde{\mathbf{x}}). \tag{2.2}$$

Note that unlike the population case, this certificate bound is probabilistic. Also note that this bound compares the empirical expectation of the saliency map at an observed point to the true population expectation at all nearby points. This convention is in line with recent works providing probabilistic certificates for smoothed classifiers (Cohen et al., 2019; Salman et al., 2019).

Theorem 1 allows us to derive certificates for the top-$K$ overlap (denoted by $R$). In particular:

**Theorem 2.** $\forall \mathbf{x}, \tilde{\mathbf{x}} \in \mathbb{R}^n$, $\|\mathbf{x} - \tilde{\mathbf{x}}\|_2 \le \rho$, $\sigma \in \mathbb{R}$, $q \in \mathbb{N}$, *define* $R^{cert}(\mathbf{x}, K)$ *as the largest* $i \le K$ *such that* $\hat{L}(\tilde{\mathbf{h}}_{[i]}(\mathbf{x})) \ge \tilde{\mathbf{h}}_{[2K-i]}(\mathbf{x})$. *Then, with probability at least* $p$,

$$R^{cert}(\mathbf{x}, K) \le R(\mathbf{x}, \tilde{\mathbf{x}}, K). \tag{2.3}$$

*where* $R(\mathbf{x}, \tilde{\mathbf{x}}, K)$ *denotes the top-K overlap between* $\tilde{\mathbf{h}}(\mathbf{x})$ *and* $\bar{\mathbf{h}}(\tilde{\mathbf{x}})$.

Intuitively, if there is a sufficiently large *gap* between the $i^{th}$ and $(2K - i)^{th}$ largest elements of an empirical smoothed saliency map, then we can certify that the overlap between top $K$ elements of the observed saliency map and any possible perturbed population smoothed saliency map is at least $i$ with probability at least $p$. If the observed saliency map is possibly adversarially corrupted up to a radius of $\rho$, it is thus still guaranteed to be similar to the "ground truth" population saliency map.

Note that it requires some modifications of SmoothGrad (Smilkov et al., 2017) for our bounds to be directly applicable. Smilkov et al. (2017) in particular defines two methods, which we will call SmoothGrad and Quadratic SmoothGrad. SmoothGrad takes the mean over samples of the signed gradient values, with absolute value typically taken after smoothing for visualization. Quadratic SmoothGrad takes the mean of the elementwise squares of gradient values. Both methods, therefore, require modification for our bounds to be applied: we define scaled SmoothGrad $\tilde{\mathbf{g}}(\mathbf{x})$, such that $\mathbf{g}(\mathbf{x})$ is the elementwise absolute value of the gradient, linearly scaled so that the largest element is one. We can similarly define a scaled Quadratic SmoothGrad. We can then apply Theorem 2 directly to scaled SmoothGrad (or scaled Quadratic SmoothGrad), simply by scaling the components of $\mathbf{g}(\mathbf{x})$ (or $\mathbf{g}(\mathbf{x}) \odot \mathbf{g}(\mathbf{x})$) to lie in the interval $[0, 1]$. However, we observe that this gives vacuous bounds for both of them when using the suggested hyperparameters from Smilkov et al. (2017). One issue is that the suggested value for $q$ (number of perturbations) is 50 which is too small to give useful bounds in Corollary 1. For a standard size image from the ImageNet dataset ($n = 224 \times 224 \times 3 = 150,528$), with $p = 0.95$, this gives $c = 0.395$ (using Definition 2.2). Note that even for a small $\rho$:

$$\hat{L}(z) = \Phi\left(\Phi^{-1}(z - c) - \frac{2\rho}{\sigma}\right) - c \approx \Phi\left(\Phi^{-1}(z - c)\right) - c = z - 2c$$

Thus the gap between $z$ and $\hat{L}(z)$ is at least $0.79$. We can see from Corollary 1 that a gap of $0.79$ (on a scale of 1) is far too large to be of any practical use. We instead take $q = 2^{13}$, which gives a more manageable estimation error of $c = 0.031$. However, we found that even with this adjustment, the bounds computed using Theorem 2 are not satisfactory for either scaled SmoothGrad and or scaled Quadratic SmoothGrad (see details in the appendix). This prompted the development of Sparsified SmoothGrad described in Section 2.3.

## 2.3 SPARSIFIED SMOOTHGRAD AND ITS RELAXATION

Scaled SmoothGrad and Quadratic SmoothGrad give vacuous robustness certificates using our proposed method, as demonstrated in Figure 1. We therefore develop a new method, *Sparsified Smooth-Grad*, which has (1) non-vacuous robustness certificates at ImageNet scale (Figure 2a), (2) similar high-quality visual output to SmoothGrad (Figure 3), and (3) theoretical guarantees that aid in setting its hyper-parameters (Section 3).

The Sparsified SmoothGrad is defined as $\tilde{\mathbf{g}}^{[\tau]}$, where $\mathbf{g}^{[\tau]}$ is defined as follows:

$$\mathbf{g}_i^{[\tau]}(\mathbf{x}) = \begin{cases} 0, & \text{if } \mathbf{g}_i(\mathbf{x}) < \mathbf{g}_{[\tau n]}(\mathbf{x}) \\ 1, & \text{if } \mathbf{g}_i(\mathbf{x}) \geq \mathbf{g}_{[\tau n]}(\mathbf{x}) \end{cases} \tag{2.4}$$

In other words, $\tau$ controls the *degree of sparsification*: a fraction $\tau$ of elements (the largest $\tau n$ elements of $\mathbf{g}(\mathbf{x})$) are assigned to 1, and the rest are set to 0.

For some applications, it may be desirable to have at least some differentiable elements in the computed saliency map. For this purpose, we also propose *Relaxed Sparsified SmoothGrad:*

$$\mathbf{g}_i^{[\gamma, \tau]}(\mathbf{x}) = \begin{cases} 0, & \text{if } \mathbf{g}_i(\mathbf{x}) < \mathbf{g}_{[\tau n]}(\mathbf{x}) \\ 1, & \text{if } \mathbf{g}_i(\mathbf{x}) \geq \mathbf{g}_{[\gamma n]}(\mathbf{x}) \\ \frac{\mathbf{g}_i(\mathbf{x})}{\mathbf{g}_{[\gamma n]}(\mathbf{x})}, & \text{otherwise} \end{cases} \tag{2.5}$$

Here, $\tau$ controls the *degree of sparsification* and $\gamma$ controls the *degree of clipping*: a fraction $\gamma$ of elements are clipped to 1. Elements neither clipped nor sparsified are linearly scaled between 0 and 1. Note that Relaxed Sparsified SmoothGrad is a generalization of Sparsified SmoothGrad. With no clipping ($\gamma = 0$), we again achieve nearly-vacuous results. However, with only a small degree of clipping ($\gamma = 0.01$), we achieve results very similar (although slightly worse) than sparsifed Smooth-Grad; see Figure 2b. We use Relaxed Sparsified SmoothGrad in this paper to test the performance of first-order adversarial attacks against Sparsified SmoothGrad-like techniques.

## 3 RANK CERTIFICATE FOR THE PROPOSED SPARSIFIED SMOOTHGRAD

In this section, we show that if the *median* rank of a saliency map element over smoothing perturbations is sufficiently small (i.e. near the top rank), then for an adversarially perturbed input, that

| Base Image | SmoothGrad | Quadratic SmoothGrad | Sparsified SmoothGrad |
|---|---|---|---|

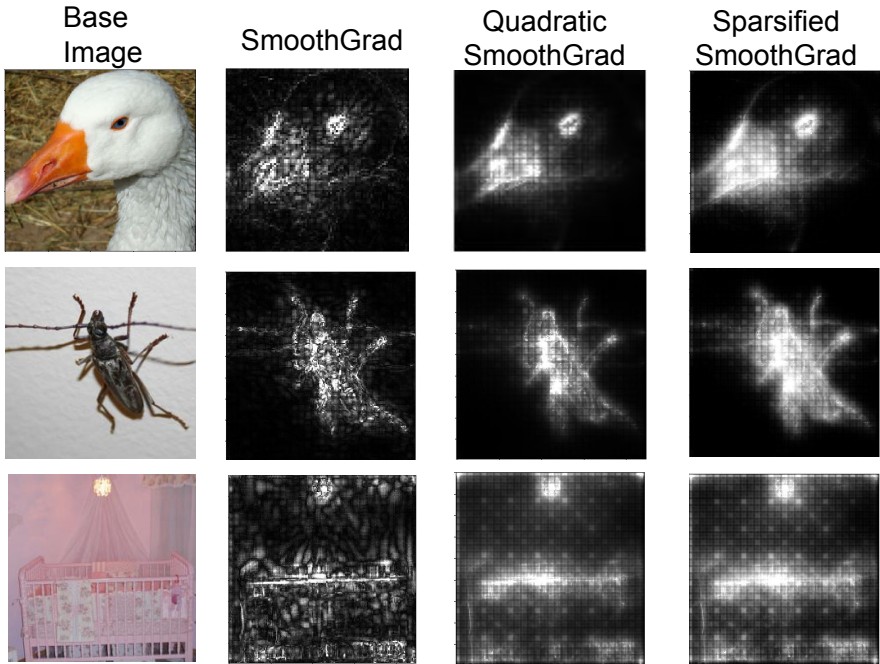

Figure 3: Qualitative comparison of Sparsified SmoothGrad (with the sparsification parameter $\tau$ = 0.1) with the SmoothGrad methods defined by Smilkov et al. (2017). All methods lead to high-quality saliency maps while our proposed Sparsified SmoothGrad is certifiably robust to adversarial examples as well. Additional examples have been presented in the appendix.

element will certifiably remain near the top rank of the proposed Sparsified SmoothGrad method with high probability. This provides a theoretical justification for the certifiable robustness of the proposed Sparsified SmoothGrad method.

To present this result, we first define the certified *rank* of an element in the saliency map as follows:

**Definition 3.1** (Certified Rank). For a given input $\mathbf{x}$ and a given saliency map method (denoted by $\mathbf{h} : \mathbb{R}^n \to \mathbb{R}^n$), let the maximum adversarial distortion be $\rho$, i.e. $\|\tilde{\mathbf{x}} - \mathbf{x}\|_2 \leq \rho$. Then, for a probability $p$, the certified rank for an element at index $i$ (denoted by $\text{rank}^{\text{cert}}(\mathbf{x}, i)$) is defined as the minimum $k$ such that the following condition holds:

$$\hat{L}(\tilde{\mathbf{h}}_i(\mathbf{x})) \geq \tilde{\mathbf{h}}_{[k]}(\mathbf{x}).$$

If the $i$-th element of the saliency map has a certified rank of $k$, using Corollary 1, we will have:

$$\bar{\mathbf{h}}_i(\tilde{\mathbf{x}}) \geq \bar{\mathbf{h}}_{[k]}(\tilde{\mathbf{x}}) \quad \text{with probability at least } p.$$

That is, the $i^{th}$ element of the population smoothed saliency map is guaranteed to be as large as the smallest $n - k + 1$ elements of the smoothed saliency map of any adversarially perturbed input.

Note that certified rank depends on the particular perturbations used to generate the smoothed saliency map $\tilde{\mathbf{h}}(\mathbf{x})$. In the following result, we show that if the median rank of a gradient element at index $i$, over a set of randomly generated perturbations, is less than a specified threshold value, then the certified rank of that element in the Sparsified SmoothGrad saliency map generated using those perturbations can be upper bounded.

**Theorem 3.** *Let $U$ be the set of $q$ random perturbations for a given input $\mathbf{x}$ using the smoothing variance $\sigma^2$. Using the Sparsified SmoothGrad method, with probability $p$, we have*

$$\underset{\epsilon \in U}{Median} \left[ \text{rank}(\mathbf{g}(\mathbf{x} + \epsilon), i) \right] \leq \lceil \tau n \rceil \quad \Rightarrow \quad \text{rank}^{cert}(\mathbf{x}, i) \leq \frac{\lceil \tau n \rceil}{\hat{L}(\frac{1}{2})}, \tag{3.1}$$

*where $\tau$ is the sparsification parameter of the Sparsified SmoothGrad method.*

For instance, if $\rho \ll \sigma$ and for sufficiently large number of smoothing perturbations (i.e. $q \to \infty$), we have $\hat{L}(1/2) \to 1/2$. If we set $\tau = K/(2n)$, then for indices whose median ranks are less than or equal to $K/2$, their certified ranks will be less than or equal to $K$. That is, even after adversarially perturbing the input, they will certifiably remain among the top $K$ elements of the Sparsified SmoothGrad saliency map. We present a more general form of this result in the appendix.

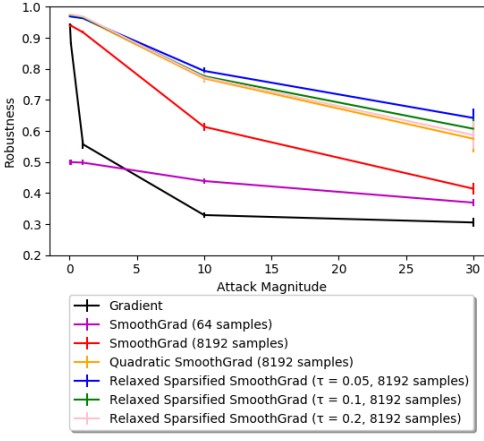

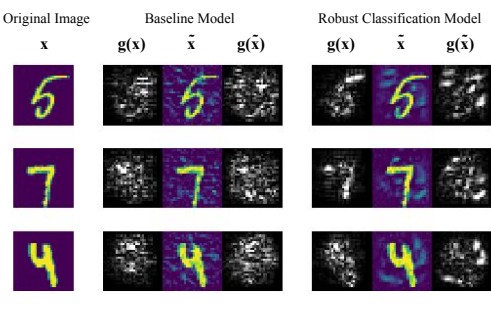

Figure 5: Adversarial training for robust classification is *not* effective at producing robust interpretations. We attack the interpretation (using an $L_2$ attack magnitude $\rho = 10$) of a baseline CNN and a CNN trained for robust classification. The classification-robust model has an average robustness of $0.5853$, compared to a baseline of $0.5727$, over the MNIST test set. Robustness here is measured as $R(\mathbf{x}, \tilde{\mathbf{x}}, K)/K$ with $K = n/4$.

Figure 4: Empirical robustness of variants of SmoothGrad to adversarial attack, tested on CIFAR-10 with ResNet-18. The attack magnitude is in units of standard deviations of pixel intensity. Robustness is measured as $R(\mathbf{x}, \tilde{\mathbf{x}}, K)/K$ with $K = n/4$.

## 4 EMPIRICAL RESULTS

To test the empirical robustness of Sparsified SmoothGrad, we used an $L_2$ attack on $R(\mathbf{x}, K)$ adapted from the $L_\infty$ attack defined by Ghorbani et al. (2019); see the appendix for details of our proposed attack. We test Relaxed Sparsified SmoothGrad ($\gamma = .01; \tau = .05, .1, .2$), rather than Sparsified SmoothGrad because our attack is gradient-based and Sparsified SmoothGrad has no defined gradients. We tested on ResNet-18 with CIFAR-10 with the attacker using a separately-trained, fully differentiable version of ResNet-18, with SoftPlus activations in place of ReLU. We present our results in Figure 4. We observe that our method is significantly more robust than the SmoothGrad method while its robustness is on par with the Quadratic SmoothGrad method with the same number of smoothing perturbations. We note that our robustness certificate appears to be loose for the large perturbation magnitudes used in these experiments; see the appendix for a direct comparison.

The attack introduced by Ghorbani et al. (2019) (and its $L_2$ extension that we introduced) does not change the the original classification of the image, so protecting against adversarial attacks on classification is an insufficient defence. To demonstrate this, we compare a model trained adversarially for *classification* with a baseline model. Specifically, we adversarially train a simple CNN classification model on MNIST (see appendix for model architecture) using a state-of-the-art $L_2$ adversarial training procedure provided by Rony et al. (2019). We used the code available for this adversarial training procedure directly. While, as predicted by Tsipras et al. (2018), the resulting model produced high-quality saliency maps, the model was not significantly more robust to adversarial perturbation tailored for the interpretation than a naively trained classifier (Figure 5, top-$K$ overlap of $58.5\%$ with adversarial training for classification vs. $57.3\%$ on the baseline). This highlights that adversarial training for classification does *not* suffice to provide adversarial robustness for interpretation. In appendix D, we propose an adversarial training approach which successfully defends interpretations against adversarial attack (top-$K$ overlap greater than $80\%$ under the same attack) while also producing high-quality saliency maps. However, unlike the proposed Sparsified SmoothGrad, this approach is not certifiably robust.

## 5 CONCLUSION

In this work, we studied the robustness of deep learning interpretation against adversarial attacks. We introduced a sparsified variant of the popular SmoothGrad method which computes the average saliency maps over random perturbations of the input. By establishing an easy-to-compute robustness certificate for the interpretation problem, we showed that the proposed Sparsified SmoothGrad is certifiably robust to adversarial attacks while producing high-quality saliency maps. We provided experiments on ImageNet samples validating our theory.

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

## A  PROOFS

**Theorem 1.** *Let* $\mathbf{h} : \mathbb{R}^n \to [0,1]^n$ *be a bounded, real-valued function,* $\sigma^2 \in \mathbb{R}$ *be the smoothing variance for* $\mathbf{h}$*, then* $\forall\ i, j \in [n]$*,* $\mathbf{x}, \tilde{\mathbf{x}} \in \mathbb{R}^n$ *where* $\tilde{\mathbf{x}} - \mathbf{x} = \delta$ *such that* $\|\delta\|_2 \le \rho$:

$$\Phi\left(\Phi^{-1}\left(\bar{\mathbf{h}}_i(\mathbf{x})\right) - \frac{2\rho}{\sigma}\right) \ge \bar{\mathbf{h}}_j(\mathbf{x}) \Rightarrow \bar{\mathbf{h}}_i(\tilde{\mathbf{x}}) \ge \bar{\mathbf{h}}_j(\tilde{\mathbf{x}})$$

*where* $\Phi$ *denotes the cdf function for the standard normal distribution and* $\Phi^{-1}$ *is its inverse.*

We will prove this by first proving a more general lemma[2]:

**Lemma 1.** *For any bounded function* $h : \mathbb{R} \to [0,1]$ *and smoothing variance* $\sigma^2 \in \mathbb{R}$*,* $\Phi^{-1}(\bar{h}(\mathbf{x}))$ *is Lipschitz-continuous with respect to* $\mathbf{x}$*, with Lipschitz constant* $\sigma^{-1}$*.*

*Proof.* By the definition of Lipschitz continuity, we must show that $\forall \delta \in \mathbb{R}^n$,

$$\Phi^{-1}(\bar{h}(\mathbf{x})) - \frac{\|\delta\|_2}{\sigma} \le \Phi^{-1}(\bar{h}(\mathbf{x}+\delta)) \le \Phi^{-1}(\bar{h}(\mathbf{x})) + \frac{\|\delta\|_2}{\sigma} \tag{A.1}$$

We first define a new, randomized function $H : \mathbb{R} \to \{0,1\}$,

$$H(\mathbf{x}) \sim \text{Bern}\left(h(\mathbf{x})\right)$$

Then $\forall\ \mathbf{x} \in \mathbb{R}^n$:

$$\mathbb{E}\left[H(\mathbf{x}+\epsilon)\right] = \mathbb{E}_\epsilon\left[\mathbb{E}_H\left[H(\mathbf{x}+\epsilon)\right]\right] = \mathbb{E}_\epsilon\left[h(\mathbf{x}+\epsilon)\right] = \bar{h}(\mathbf{x}) \tag{A.2}$$

Now, we apply the following Lemma (Lemma 4 from Cohen et al. (2019)):

**Lemma** (Lemma 4 from (Cohen et al., 2019)). *Let* $X \sim \mathcal{N}(\mathbf{x},\ \sigma^2 I)$ *and* $Y \sim \mathcal{N}(\mathbf{x}+\delta,\ \sigma^2 I)$*. Let* $f : \mathbb{R}^n \to \{0,1\}$ *be any deterministic or random function, Then:*

1. *If* $S = \left\{z \in \mathbb{R}^n : \delta^T z \le \beta\right\}$ *for some* $\beta$ *and* $\Pr(f(X) = 1) \ge \Pr(X \in S)$*, then* $\Pr(f(Y) = 1) \ge \Pr(Y \in S)$

2. *If* $S = \left\{z \in \mathbb{R}^n : \delta^T z \ge \beta\right\}$ *for some* $\beta$ *and* $\Pr(f(X) = 1) \le \Pr(X \in S)$*, then* $\Pr(f(Y) = 1) \le \Pr(Y \in S)$

Using the same technique as used in the proof of Theorem 1 in Cohen et al. (2019), we fix $\mathbf{x}, \delta$ and define,

$$\beta = \sigma\|\delta\|_2\Phi^{-1}(\mathbb{E}\left[H(\mathbf{x}+\epsilon)\right])$$

Also define the half-spaces:

$$S_- = \left\{\mathbf{z} : \delta^T\mathbf{z} \le \beta + \delta^T\mathbf{x}\right\} = \left\{z : \delta^T(\mathbf{z}-\mathbf{x}) \le \beta\right\}$$
$$S_+ = \left\{\mathbf{z} : \delta^T\mathbf{z} \ge -\beta + \delta^T\mathbf{x}\right\} = \left\{z : \delta^T(\mathbf{z}-\mathbf{x}) \ge -\beta\right\}$$

Applying algebra from the proof of Theorem 1 in Cohen et al. (2019), we have,

$$\Pr(X \in S_-) = \Phi\left(\frac{\beta}{\sigma\|\delta\|_2}\right) = \mathbb{E}\left[H(\mathbf{x}+\epsilon)\right] \tag{A.3}$$

$$\Pr(X \in S_+) = 1 - \Phi\left(\frac{-\beta}{\sigma\|\delta\|_2}\right) = 1 - \left(1 - \Phi\left(\frac{\beta}{\sigma\|\delta\|_2}\right)\right) = \mathbb{E}\left[H(\mathbf{x}+\epsilon)\right] \tag{A.4}$$

$$\Pr(Y \in S_-) = \Phi\left(\frac{\beta}{\sigma\|\delta\|_2} - \frac{\|\delta\|_2}{\sigma}\right) = \Phi\left(\Phi^{-1}(\mathbb{E}\left[H(\mathbf{x}+\epsilon)\right]) - \frac{\|\delta\|_2}{\sigma}\right) \tag{A.5}$$

$$\Pr(Y \in S_+) = \Phi\left(\frac{-(-\beta)}{\sigma\|\delta\|_2} + \frac{\|\delta\|_2}{\sigma}\right) = \Phi\left(\Phi^{-1}(\mathbb{E}\left[H(\mathbf{x}+\epsilon)\right]) + \frac{\|\delta\|_2}{\sigma}\right) \tag{A.6}$$

---

[2]We note that Salman et al. (2019) has recently and independently published a different proof of this lemma.

Using equation A.3

$$\Pr(H(X) = 1) = \mathbb{E}\left[H(X)\right] = \mathbb{E}\left[H(\mathbf{x} + \epsilon)\right] \geq \Pr(X \in S_-)$$

Applying Statement 1 of Cohen's lemma, using $f = H$ and $S = S_-$:

$$\mathbb{E}\left[H(\mathbf{x} + \delta + \epsilon)\right] = \Pr(H(\mathbf{x} + \delta + \epsilon) = 1) = \Pr(H(Y) = 1) \geq \Pr(Y \in S_-) \tag{A.7}$$

Using equation A.4,

$$\Pr(H(X) = 1) = \mathbb{E}\left[H(X)\right] = \mathbb{E}\left[H(\mathbf{x} + \epsilon)\right] \leq \Pr(X \in S_+)$$

Applying Statement 2 of Cohen's lemma, using $f = H$ and $S = S_+$:

$$\mathbb{E}\left[H(\mathbf{x} + \delta + \epsilon)\right] = \Pr(H(\mathbf{x} + \delta + \epsilon) = 1) = \Pr(H(Y) = 1) \leq \Pr(Y \in S_+) \tag{A.8}$$

Using equation A.7 and equation A.8:

$$\Pr(Y \in S_-) \leq \mathbb{E}\left[H(\mathbf{x} + \delta + \epsilon)\right] \leq \Pr(Y \in S_+)$$

Then by equation A.5 and equation A.6:

$$\Phi\left(\Phi^{-1}(\mathbb{E}\left[H(\mathbf{x} + \epsilon)\right]) - \frac{\|\delta\|_2}{\sigma}\right) \leq \mathbb{E}\left[H(\mathbf{x} + \delta + \epsilon)\right] \leq \Phi\left(\Phi^{-1}(\mathbb{E}\left[H(\mathbf{x} + \epsilon)\right]) + \frac{\|\delta\|_2}{\sigma}\right)$$

Noting that $\Phi^{-1}$ is a monotonically increasing function, we have:

$$\Phi^{-1}(\mathbb{E}\left[H(\mathbf{x} + \epsilon)\right]) - \frac{\|\delta\|_2}{\sigma} \leq \Phi^{-1}(\mathbb{E}\left[H(\mathbf{x} + \delta + \epsilon)\right]) \leq \Phi^{-1}(\mathbb{E}\left[H(\mathbf{x} + \epsilon)\right]) + \frac{\|\delta\|_2}{\sigma}$$

Using equation A.2 yields equation A.1, which completes the proof. $\qquad\square$

We now proceed with the proof of Theorem 1.

*Proof.* Applying Lemma 1 to $\mathbf{h}_i$ and $\mathbf{h}_j$ gives (recalling that $\tilde{\mathbf{x}} = \delta + \mathbf{x}$ ):

$$\Phi^{-1}(\bar{\mathbf{h}}_i(\mathbf{x})) - \frac{\rho}{\sigma} \leq \Phi^{-1}(\bar{\mathbf{h}}_i(\mathbf{x})) - \frac{\|\delta\|_2}{\sigma} \leq \Phi^{-1}(\bar{\mathbf{h}}_i(\tilde{\mathbf{x}})) \tag{A.9}$$

$$\Phi^{-1}(\bar{\mathbf{h}}_j(\tilde{\mathbf{x}})) \leq \Phi^{-1}(\bar{\mathbf{h}}_j(\mathbf{x})) + \frac{\|\delta\|_2}{\sigma} \leq \Phi^{-1}(\bar{\mathbf{h}}_j(\mathbf{x})) + \frac{\rho}{\sigma} \tag{A.10}$$

Then we have:

$$\Phi\left(\Phi^{-1}\left(\bar{\mathbf{h}}_i(\mathbf{x})\right) - \frac{2\rho}{\sigma}\right) \geq \bar{\mathbf{h}}_j(\mathbf{x})$$

$$\implies \quad \Phi^{-1}\left(\bar{\mathbf{h}}_i(\mathbf{x})\right) - \frac{\rho}{\sigma} \geq \Phi^{-1}\left(\bar{\mathbf{h}}_j(\mathbf{x})\right) + \frac{\rho}{\sigma} \quad \text{(by monotonicity of } \Phi^{-1}\text{)}$$

$$\implies \quad \Phi^{-1}(\bar{\mathbf{h}}_i(\tilde{\mathbf{x}})) \geq \Phi^{-1}\left(\bar{\mathbf{h}}_j(\mathbf{x})\right) + \frac{\rho}{\sigma} \quad \text{(by equation A.9 )}$$

$$\implies \quad \Phi^{-1}(\bar{\mathbf{h}}_i(\tilde{\mathbf{x}})) \geq \Phi^{-1}(\bar{\mathbf{h}}_j(\tilde{\mathbf{x}})) \quad \text{(by equation A.10 )}$$

$$\implies \quad \bar{\mathbf{h}}_i(\tilde{\mathbf{x}}) \geq \bar{\mathbf{h}}_j(\tilde{\mathbf{x}}) \quad \text{(by monotonicity of } \Phi\text{)}$$

which proves the implication. $\qquad\square$

**Corollary 1.** *Let* $\mathbf{h} : \mathbb{R}^n \to [0,1]^n$ *be a function such that for given values of* $q$, $\sigma$:

$$\tilde{\mathbf{h}}(\mathbf{x}) = \frac{1}{q} \sum_{i=1}^{q} \mathbf{h}(\mathbf{x} + \epsilon_i), \quad \epsilon_i \sim N(0, \sigma^2 I) \tag{A.11}$$

$\forall \, i, j \in [n]$, $\mathbf{x}$, $\tilde{\mathbf{x}} \in \mathbb{R}^n$, $\|\mathbf{x} - \tilde{\mathbf{x}}\|_2 \leq \rho$, *with probability at least* $p$,

$$\hat{L}(\tilde{\mathbf{h}}_i(\mathbf{x})) \geq \tilde{\mathbf{h}}_j(\mathbf{x}) \Rightarrow \bar{\mathbf{h}}_i(\tilde{\mathbf{x}}) \geq \bar{\mathbf{h}}_j(\tilde{\mathbf{x}})$$

*Proof.* By Hoeffding's Inequality, for any $c > 0$, $\forall\, i$ :

$$\Pr\left[|\tilde{\mathbf{h}}_i(\mathbf{x}) - \bar{\mathbf{h}}_i(\mathbf{x})| \ge c\right] \le 2e^{-2qc^2} \tag{A.12}$$

Then:

$$\Pr\left[\bigcup_i\left(|\tilde{\mathbf{h}}_i(\mathbf{x}) - \bar{\mathbf{h}}_i(\mathbf{x})| \ge c\right)\right] \le 2ne^{-2qc^2} \tag{A.13}$$

Since we are free to choose c, we define $c$ such that $1 - p = 2ne^{-2qc^2}$, then:

$$c = \sqrt{\frac{ln(2n(1-p)^{-1})}{2q}} \tag{A.14}$$

$$\Pr\left[\bigcup_i\left(|\tilde{\mathbf{h}}_i(\mathbf{x}) - \bar{\mathbf{h}}_i(\mathbf{x})| \ge c\right)\right] \le 2ne^{-2qc^2} = 1 - p$$

$$\implies 1 - \Pr\left[\bigcup_i\left(|\tilde{\mathbf{h}}_i(\mathbf{x}) - \bar{\mathbf{h}}_i(\mathbf{x})| \ge c\right)\right] \ge p$$

$$\implies \Pr\left[\bigcap_i\left(|\tilde{\mathbf{h}}_i(\mathbf{x}) - \bar{\mathbf{h}}_i(\mathbf{x})| < c\right)\right] \ge p$$

Then with probability at least $p$:

$$\begin{aligned}
\tilde{\mathbf{h}}_i(\mathbf{x}) - c &< \bar{\mathbf{h}}_i(\mathbf{x}) \\
\tilde{\mathbf{h}}_j(\mathbf{x}) + c &> \bar{\mathbf{h}}_j(\mathbf{x})
\end{aligned} \tag{A.15}$$

So:

$$\Phi\left(\Phi^{-1}\left(\tilde{\mathbf{h}}_i(\mathbf{x}) - c\right)\right) - \frac{2\rho}{\sigma} \ge \tilde{\mathbf{h}}_j(\mathbf{x}) + c \implies \Phi\left(\Phi^{-1}\left(\bar{\mathbf{h}}_i(\mathbf{x})\right)\right) - \frac{2\rho}{\sigma} \ge \bar{\mathbf{h}}_j(\mathbf{x}) \tag{A.16}$$

The result directly follows from Theorem 1. $\qquad\square$

**Theorem 2.** $\forall\, \mathbf{x}, \tilde{\mathbf{x}} \in \mathbb{R}^n$, $\|\mathbf{x} - \tilde{\mathbf{x}}\|_2 \le \rho$, $\sigma \in \mathbb{R}$, $q \in \mathbb{N}$, define $R^{cert}(\mathbf{x}, K)$ as the largest $i \le K$ such that $\hat{L}(\tilde{\mathbf{h}}_{[i]}(\mathbf{x})) \ge \tilde{\mathbf{h}}_{[2K-i]}(\mathbf{x})$. Then, with probability at least p,

$$R^{cert}(\mathbf{x},\ K) \le R(\mathbf{x},\ \tilde{\mathbf{x}},\ K). \tag{A.17}$$

*where $R(\mathbf{x},\ \tilde{\mathbf{x}},\ K)$ denotes the top-K overlap between $\tilde{\mathbf{h}}(\mathbf{x})$ and $\bar{\mathbf{h}}(\tilde{\mathbf{x}})$.*

*Proof.* Note that the proof of Corollary 1 guarantees that with probability at least $p$, *all* estimates $\tilde{\mathbf{h}}(\mathbf{x})$ are within the approximation bound $c$ of $\bar{\mathbf{h}}(\mathbf{x})$. So we can assume that Corollary 1 will apply simultaneously to all pairs of indices $i, j$, with probability $p$.
We proceed to prove by contradiction.

$$\text{Let } i = R^{\text{cert}}(\mathbf{x}, K)$$

$$\implies \hat{L}(\tilde{\mathbf{h}}_{[i]}(\mathbf{x})) \ge \tilde{\mathbf{h}}_{[2K-i]}(\mathbf{x}),$$

Suppose there exists $\tilde{\mathbf{x}}$ such that:

$$R(\mathbf{x},\ \tilde{\mathbf{x}},\ K) < i,$$

Since $\hat{L}$ is a monotonically increasing function,

$$\hat{L}(\tilde{\mathbf{h}}_{[i]}(\mathbf{x})) \ge \tilde{\mathbf{h}}_{[2K-i]}(\mathbf{x})$$

$$\implies \hat{L}(\tilde{\mathbf{h}}_{[i']}(\mathbf{x})) \ge \tilde{\mathbf{h}}_{[j']}(\mathbf{x}), \quad \forall\, i' \le i,\ j' \ge 2K - i,$$

and therefore by Corollary 1:

$$\forall\, m, n \quad \text{rank}(\tilde{\mathbf{h}}(\mathbf{x}), m) \le i,\ \text{rank}(\tilde{\mathbf{h}}(\mathbf{x}), n) \ge 2K - i \implies \bar{\mathbf{h}}_m(\tilde{\mathbf{x}})) \ge \bar{\mathbf{h}}_n(\tilde{\mathbf{x}}) \tag{A.18}$$

Let $X$ be the set of indices in the top $K$ elements in $\tilde{\mathbf{h}}(\mathbf{x})$, and $\tilde{X}$ be the set of indices in the top $K$ elements in $\bar{\mathbf{h}}(\tilde{\mathbf{x}})$.
By assumption, $X$ and $\tilde{X}$ share fewer than $i$ elements, so there will be at least $K - i + 1$ elements in

$\tilde{X}$ which are not in $X$.

All of these elements have rank at least $K + 1$ in $\tilde{\mathbf{h}}(\mathbf{x})$.

Thus by pigeonhole principle, there is some index $l \in \tilde{X} - X$, such that $\operatorname{rank}(\tilde{\mathbf{h}}(\mathbf{x}), l) \geq K + K - i + 1 = 2K - i + 1 \geq 2K - i$.

Thus by Equation equation A.18,

$$\forall m, \text{ where } \operatorname{rank}(\tilde{\mathbf{h}}(\mathbf{x}), m) \leq i, \quad \bar{\mathbf{h}}_m(\tilde{\mathbf{x}}) \geq \bar{\mathbf{h}}_l(\tilde{\mathbf{x}}) \tag{A.19}$$

Hence, there are $i$ such elements where $\operatorname{rank}(\tilde{\mathbf{h}}(\mathbf{x}), m) \leq i$: these elements are clearly in $X$.

Because $l \in \tilde{X}$, Equation equation A.19 implies that these elements are all also in $\tilde{X}$. Thus $X$ and $\tilde{X}$ share at least $i$ elements, which contradicts the premise.

(In this proof we have implicitly assumed that the top $K$ elements of a vector can contain more than $K$ elements, if ties occur, but that $\operatorname{rank}$ is assigned arbitrarily in cases of ties. In practice, ties in smoothed scores will be very unlikely.) $\qquad \square$

## A.1 GENERAL FORM AND PROOF OF THEOREM 3

We note that Theorem 3 can be used to derive a more general bound for any saliency map method that for an input $\mathbf{x}$, first maps $\mathbf{g}(\mathbf{x})$ to an elementwise function that only depends on the rank of the current element in $\mathbf{g}(\mathbf{x})$ and not on the individual value of the element. We denote the composition of the gradient function and this elementwise function as $\mathbf{g}^{[\text{rank}]}$. The only properties that the function must satisfy is that it must be monotonically decreasing and non-negative. Thus, we have the following statement:

**Theorem 3.** *Let $T$ be the threshold value and let $U$ be the set of $q$ random perturbations for a given input $\mathbf{x}$ using the smoothing variance $\sigma^2$ and let $p$ be the probability bound. If $i$ is an element index such that:*

$$\underset{\epsilon \in U}{\text{Median}} \left[ \operatorname{rank}(\mathbf{g}(\mathbf{x} + \epsilon), i) \right] \leq T \tag{A.20}$$

*Then:*

$$\operatorname{rank}^{cert}(\mathbf{x}, \, i) \leq \frac{\sum_{j=1}^{n} \mathbf{g}_{[j]}^{[\text{rank}]}(\mathbf{x})}{\hat{L}\left( \frac{\mathbf{g}_{[T]}^{[\text{rank}]}(\mathbf{x})}{2} \right)} \tag{A.21}$$

*Furthermore:*

$$\sum_{j=1}^{n} \mathbf{g}_{[j]}^{[\text{rank}]}(\mathbf{x}), \; \hat{L}\left( \frac{\mathbf{g}_{[T]}^{[\text{rank}]}(\mathbf{x})}{2} \right) \text{ are both independent of } \mathbf{x}. \text{ Thus RHS is a constant.} \tag{A.22}$$

*Proof.* Let the elementwise function be $f : \mathbb{N} \to \mathbb{R}^+$, i.e $f$ takes the rank of the element as the input and outputs a real number. Furthermore, we assume that $f$ is a non-negative monotonically decreasing function. Thus $\mathbf{g}_i^{[\text{rank}]}(\mathbf{x}) = f(\operatorname{rank}(\mathbf{g}(\mathbf{x}), i))$.

We use $f(i)$ to denote the constant value that $f$ maps elements of rank $i$ to.

Note that $\mathbf{g}_{[i]}^{[\text{rank}]}(\mathbf{x})$ is the $i^{th}$ largest element of $\mathbf{g}^{[\text{rank}]}(\mathbf{x})$.

Since $f$ is a monotonically decreasing function:

$$\mathbf{g}_{[i]}^{[\text{rank}]}(\mathbf{x}) = f(i) \quad \forall \, i \in [n]$$

Thus $\mathbf{g}_{[i]}^{[\text{rank}]}(\mathbf{x})$ is independent of $\mathbf{x}$, we simply use $\mathbf{g}_{[i]}^{[\text{rank}]}(\cdot)$ to denote $f(i)$, i.e:

$$\mathbf{g}_{[i]}^{[\text{rank}]}(\cdot) = f(i) \quad \forall \, i \in [n]$$

Because $\text{Median}_{\epsilon \in U} \left[ \operatorname{rank}(\mathbf{g}(\mathbf{x} + \epsilon), i) \right] \leq T$, for at least half of sampling instances $\epsilon$ in $U$, $\operatorname{rank}(\mathbf{g}(\mathbf{x} + \epsilon), i) \leq T$.

So in these instances $\mathbf{g}_i^{[\text{rank}]}(\mathbf{x} + \epsilon) \geq f(T)$,

The remaining half or fewer elements are mapped to other nonnegative values.

Thus the sample mean:

$$\tilde{\mathbf{g}}_i^{[\text{rank}]}(\mathbf{x}) = \frac{1}{q} \sum_{\epsilon \in U} \mathbf{g}_i^{[\text{rank}]}(\mathbf{x} + \epsilon) \geq \mathbf{g}_{[T]}^{[\text{rank}]}(\cdot)/2$$

Using Corollary 1, $\bar{\mathbf{g}}_i^{[rank]}(\mathbf{x})$ is certifiably as large as all elements with indices j such that:

$$\hat{L}(\mathbf{g}_{[T]}^{[\text{rank}]}(\cdot)/2) \geq \tilde{\mathbf{g}}_j^{[rank]}(\mathbf{x})$$

.

Now we will find an upper bound on the number of elements with indices j such that:

$$\tilde{\mathbf{g}}_j^{[rank]}(\mathbf{x}) > \hat{L}(\mathbf{g}_{[T]}^{[\text{rank}]}(\cdot)/2)$$

Because all the ranks from 1 to $n$ will occur in every sample in U, we have:

$$\forall\, \epsilon \in U, \quad \sum_{k=1}^{n} \mathbf{g}_k^{[\text{rank}]}(\mathbf{x}+\epsilon) = \sum_{k=1}^{n} \mathbf{g}_{[k]}^{[\text{rank}]}(\cdot)$$

$$\implies \sum_{k=1}^{n} \tilde{\mathbf{g}}_k^{[\text{rank}]}(\mathbf{x}) = \sum_{k=1}^{n} \frac{1}{q} \sum_{\epsilon \in U} \mathbf{g}_i^{[\text{rank}]}(\mathbf{x}+\epsilon) = \sum_{k=1}^{n} \mathbf{g}_{[k]}^{[\text{rank}]}(\cdot)$$

Thus strictly fewer than $\sum_{k=1}^{n} \mathbf{g}_{[k]}^{[\text{rank}]}(\cdot)/\hat{L}\left(\mathbf{g}_{[T]}^{[\text{rank}]}(\cdot)/2\right)$ elements will have mean greater than $\hat{L}\left(\mathbf{g}_{[T]}^{[\text{rank}]}(\cdot)/2\right)$.

Hence, $\bar{\mathbf{g}}_i(\mathbf{x})$ is certifiably at least as large as $n - \left(\sum_{k=1}^{n} \mathbf{g}_{[k]}^{[\text{rank}]}(\cdot)/\hat{L}(\mathbf{g}_{[T]}^{[\text{rank}]}(\cdot)/2)\right) + 1$ elements, which by the definition of $\text{rank}^{\text{cert}}(\mathbf{x}, i)$ yields the result.

Theorem 3 in the main text follows trivially, because in the Sparsified SmoothGrad case, $\sum_{k=1}^{n} \mathbf{g}_{[k]}^{[\tau]}(\cdot) = T$, and $\mathbf{g}_{[T]}^{[\tau]}(\cdot) = 1$. Note that this represents the tightest possible realization of this general theorem. □

## B  RELATED WORKS

Sundararajan et al. (2017) defines a baseline, which represents an input absent of information and determines feature importance by accumulating gradient information along the path from the baseline to the original input. Alvarez-Melis & Jaakkola (2018) builds interpretable neural networks by learning basis concepts that satisfy an interpretability criteria. Adebayo et al. (2018) proposes methods to assess the quality of saliency maps. Although these methods can produce visually pleasing results, they can be sensitive to noise and adversarial perturbations.

Szegedy et al. (2014) introduced adversarial attacks for classification in deep learning. That work dealt with $L_2$ attacks, and uses L-BFGS optimization to minimize the norm of the perturbation. Carlini & Wagner (2017) provide an $L_2$ attack for classification which is often considered state of the art.

One strategy to make classifiers more robust to adversarial attacks is randomized smoothing. Lecuyer et al. (2019) use randomized smoothing to develop certifiably robust classifiers in both the $L_1$ and $L_2$ norms. They show that if Gaussian smoothing is applied to class scores, a gap between the highest smoothed class score and the next highest smoothed score implies that the highest smoothed class score will still be highest under all perturbations of some magnitude. This guarantees that the smoothed classifier will be robust under adversarial perturbation.

Li et al. (2018) and Cohen et al. (2019) consider a related formulation. Cohen gives a bound that is tight in the case of linear classifiers and gives significantly larger certified radii. In their formulation, the unsmoothed classifier $c$ is treated as a black box outputting just a discrete class label. The smoothed classifier outputs the class observed with greatest frequency over noisy samples. Salman et al. (2019) considers adversarial attacks against smoothed classifiers.

In the last couple of years, several approaches have been proposed to for interpreting neural network outputs. Simonyan et al. (2014) computes the gradient of the class score with respect to the input. Smilkov et al. (2017) computes the average gradient-based importance values generated from several noisy versions of the input. Sundararajan et al. (2017) defines a baseline, which represents an

input absent of information and determines feature importance by accumulating gradient information along the path from the baseline to the original input. Alvarez-Melis & Jaakkola (2018) builds interpretable neural networks by learning basis concepts that satisfy an interpretability criteria. Adebayo et al. (2018) proposes methods to assess the quality of saliency maps. Although these methods can produce visually pleasing results, they can be sensitive to noise and adversarial perturbations (Ghorbani et al., 2019; Kindermans et al., 2019).

As mentioned in Section 1, several approaches have been introduced for interpreting image classification by neural networks (Simonyan et al., 2014; Smilkov et al., 2017; Sundararajan et al., 2017; Shrikumar et al., 2017).

## C $L_2$ ATTACK ON SALIENCY MAPS

We developed an $L_2$ norm attack on $R^{\text{cert}}$, based on Ghorbani et al. (2019)'s $L_\infty$ attack. Our algorithm is presented as Algorithm 1. We deviate from Ghorbani et al. (2019)'s attack in the following ways.:

- We use gradient descent, rather than gradient sign descent: this is a direct adaptation to the $L_2$ norm.

- We initialize learning rate as $\frac{\rho}{\|\nabla D(\mathbf{x}^0)\|_2}$, and then decrease learning rate with increasing iteration count, proportionately (for the most part) to the reciprocal of the iteration count. These are both standard practices for gradient descent.

- We use random initialization and random restarts, also standard optimization practices.

- If a gradient descent step would cross a decision boundary, we use backtracking line search to reduce the learning rate until the step stays on the correct-class side. This allows the optimization to get arbitrarily close to decision boundaries without crossing them.

We measured the effectiveness of our attack ($Q = 100, P = 20, T = 5$) against a slight modification of Ghorbani et al. (2019)'s attack, in which the image was projected (if necessary) onto the $L_2$ ball at every iteration, and also clipped to fit within image box constraints (this was not mentioned in Ghorbani et al. (2019)'s original algorithm). For this attack, we set the ($L_\infty$) learning rate parameter at $\rho/500$, and ran for up to 100 iterations. We also tested against random perturbations. For random perturbations, up to 100 points were tested until a point in the correct class was identified. We tested these attacks on both "vanilla gradient" and SmoothGrad saliency maps. See Figure 6. Experimental conditions are as described in Section F for experiments on CIFAR-10. In this figure, for each attack magnitude, we discard any image on which any optimization method failed.

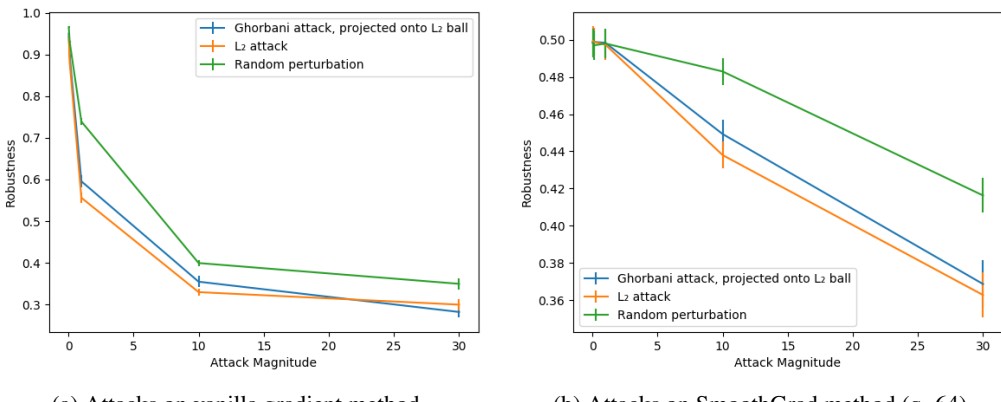

(a) Attacks on vanilla gradient method.    (b) Attacks on SmoothGrad method (q=64).

Figure 6: Comparison of attack methods on images in CIFAR-10. See text of section C.

---

**Algorithm 1** $L_2$ attack on *top-k overlap*

---

**Input:** $k$, image $\mathbf{x}$, saliency map function $\mathbf{h}$, iteration number $P$, random sampling iteration number $Q$, $L_2$ perturbation norm constraint $\rho$, classifier $c$, restarts number $T$

**Output:** Adversarial example $\tilde{\mathbf{x}}$

1: Define $D(\mathbf{z}) = -\sum_{i,\,\mathrm{rank}(\mathbf{h}(\mathbf{x}),i)\leq k} \mathbf{h}_i(\mathbf{z})$
2: **for** $t = 1, ..., T$ **do**
3:     **loop**
4:         $\delta \leftarrow$ Uniformly random vector on $L_2 = \rho$ sphere.
5:         $\mathbf{x}^0 \leftarrow \mathbf{x} + \delta$
6:         Clip $\mathbf{x}^0$ such that it falls within image box constraints.
7:         **if** $c(\mathbf{x}^0) = c(\mathbf{x})$ **then break inner loop**
8:         **if** $Q$ total iterations have passed over all $t$ cycles of random sampling **then**
9:            **break outer loop**
10:         **end if**
11:     **end loop**
12:     $\alpha \leftarrow \frac{\rho}{\|\nabla D(\mathbf{x}^0)\|_2}$
13:     **for** $p = 1, ..., P$ **do**
14:         **loop**
15:            $\mathbf{x}^p \leftarrow \mathbf{x}^{p-1} + \alpha \nabla D(\mathbf{x}^{p-1})$
16:            If necessary, project $\mathbf{x}^p$ such that $\|\mathbf{x}^p - \mathbf{x}\|_2 \leq \rho$
17:            Clip $\mathbf{x}^p$ such that it falls within image box constraints.
18:            **if** $c(\mathbf{x}^p) = c(\mathbf{x})$ **then**
19:                **break inner loop**
20:            **else**
21:                $\alpha \leftarrow \frac{\alpha}{2}$
22:            **end if**
23:         **end loop**
24:         $\alpha \leftarrow \frac{p\alpha}{p+1}$
25:     **end for**
26:     $\tilde{\mathbf{x}}^t = \arg\max_{\mathbf{z}\in\{\mathbf{x}^1,...,\mathbf{x}^P\}} D(\mathbf{z})$
27: **end for**
28: **if** random sampling failed at every iteration **then fail**
29: $\tilde{\mathbf{x}} = \arg\max_{\mathbf{z}\in\{\tilde{\mathbf{x}}^1,...,\tilde{\mathbf{x}}^T\}} D(\mathbf{z})$

---

## D    ADVERSARIAL TRAINING FOR ROBUST SALIENCY MAPS

Adversarial training has been used extensively for making neural networks robust against adversarial attacks on classification (Madry et al., 2018). The key idea is to generate adversarial examples for a classification model, and then re-train the model on these adversarial examples.

In this section, we present an adversarial training approach for fortifying deep learning interpretations so that the saliency maps generated by the model (during test time) are robust against adversarial examples. We focus on "vanilla gradient" saliency maps, although the technique presented here can potentially be applied to any saliency map method which is differentiable w.r.t. the input. We solve the following optimization problem for the network weights (denoted by $\theta$):

$$\min_{\theta} \quad \mathbb{E}_{(\mathbf{x},y)\sim D}\left[ \underbrace{\ell_{\mathrm{cls}}(\mathbf{x}, y)}_{\text{Classification loss}} + \underbrace{\lambda \|\mathbf{g}(\mathbf{x}) - \mathbf{g}(\tilde{\mathbf{x}})\|_2^2}_{\text{Robustness loss}} \right], \tag{D.1}$$

where $\tilde{\mathbf{x}}$ is an adversarial perturbation for the saliency map generated from $\mathbf{x}$. To generate $\tilde{\mathbf{x}}$, we developed an $L_2$ attack on saliency maps by extending the $L_\infty$ attack of Ghorbani et al. (2019) (see the details in Section C). $\ell_{\mathrm{cls}}(\mathbf{x}, y)$ is the standard cross entropy loss, and $\lambda$ is the regularization parameter to encourage consistency between saliency maps of the original and adversarially perturbed images.

We observe that the proposed adversarial training significantly improves the robustness of saliency maps. Aggregate empirical results are presented in Figure 7, and examples of saliency maps are presented in Figure 8. It is notable that the quality of the saliency maps is greatly improved for

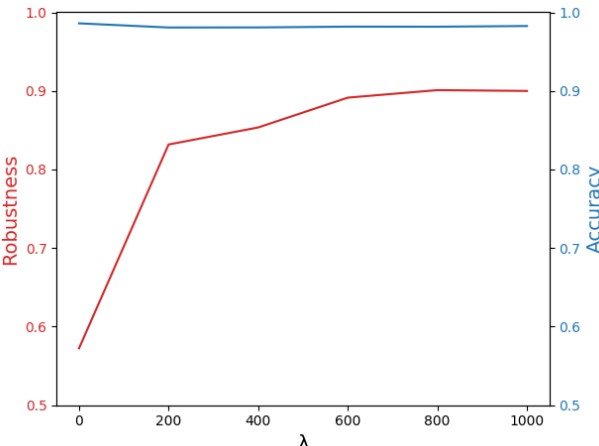

Figure 7: Effectiveness of adversarial training on MNIST. Increasing the regularization parameter $\lambda$ in the proposed adversarial training optimization (Equation D.1) significantly increases the robustness of gradient-based saliency maps while it has little effect on the classification accuracy. Robustness here is measured as $R(\mathbf{x}, \tilde{\mathbf{x}}, K)/K$, where $K = n/4$.

.

unperturbed inputs, by adversarial training. We note that this observation is related to the observation made in Tsipras et al. (2018) showing that adversarial training for classification improves the quality of the gradient-based saliency maps. We observe that even for very large value of $\lambda$, only a slight reduction in classification accuracy occurs due to the added regularization term.

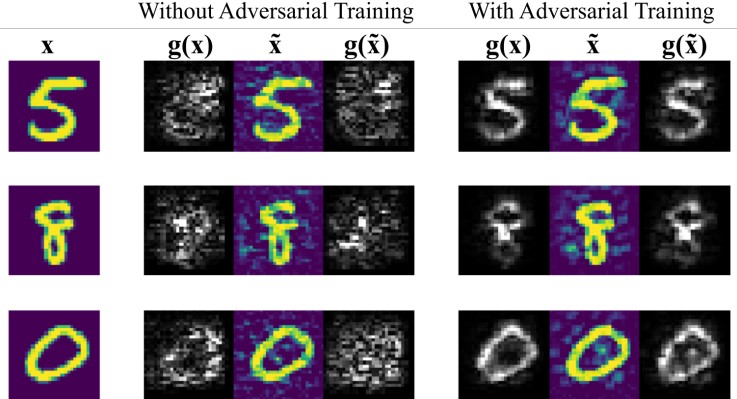

Figure 8: An illustration of the proposed adversarial training to robustify deep learning interpretation on MNIST. We observe that the proposed adversarial training not only enhances the robustness but it also improves the quality of the gradient-based saliency maps.

## E   ADVERSARIAL TRAINING ARCHITECTURE DETAILS

We use the Adam optimizer and generate new adversarial examples after each batch of training according to the updated model. We use a simple convolutional neural network, with SoftPlus activations to ensure differentiability of the saliency map, on the MNIST data set (Figure 9). Adversarial perturbations of norm up to $\rho = 10$ standard deviations of pixel intensity were used. The adversarial attack used was the $L_2$ attack described in Algorithm 1, with $P = 15, Q = 100, T = 3$. Training was performed for 30 epochs using 48,000 images from the MNIST training set, testing was on the entire MNIST test set. Instances where Algorithm 1 failed were not counted in the averages of

saliency map robustness, and were rare. (Highest frequency was for $\lambda = 0$, at 0.11%). We used the implementation of Adam Optimizer provided with PyTorch (Paszke et al., 2017), with default training parameters. These are (Table 2):

**Table 1** Hyper-parameters used in model training for MNIST experiments.

| | |
|---|---|
| Learning Rate | 0.001 |
| $L_2$ regularization parameter | 0 |
| $\beta$ | (.9,.999) |
| $\epsilon$ | $10^{-8}$ |
| Batch Size | 512 |

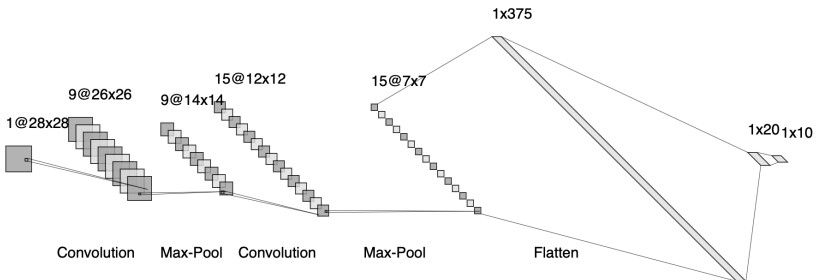

Figure 9: Network architecture used for the MNIST classification. SoftPlus activations are applied after both convolutional layers, and after the first fully connected layer.

## F  DESCRIPTION OF EXPERIMENTS IN MAIN TEXT

For ImageNet experiments (Figures 1-b and 2-a,b), we use ResNet-50, using the model pre-trained on ImageNet that is provided by *torchvision.models*, and images were pre-processed according to the recommended procedure for that model. In all of these figures, data are from the $ILSVRC2012$ validation set, sample size is 64, and the main data lines represent the $60^{th}$ percentile in the sample of the calculated robustness certificate. Error bars represent the $48^{th}$ and $72^{th}$ percentile values, corresponding to a 95% confidence interval for the population quantile.

For CIFAR-10 experiments, we train a ResNet-18 model on the CIFAR-10 training set (with pixel intensities normalized to $\sigma = 1, \mu = 0$ in each channel) using Stochastic Gradient Descent with Momentum as implemented by PyTorch (Paszke et al., 2017). The following training parameters were used:

**Table 2** Hyper-parameters used in model training for CIFAR experiments.

| | |
|---|---|
| Momentum | 0.9 |
| $L_2$ regularization parameter | .0005 |
| Epochs (max) | 375 |
| Learning Rate Schedule | .1 (epoch < 150), .01 (epoch $\geq$ 150) |
| Batch Size | 128 |

Twenty percent of the training data was used for validation, and early stopping was used to maximize accuracy relative to this validation set. For adversarial attacks on CIFAR samples, we train a version of ResNet-18 with SoftMax activations instead of ReLU. The adversarial attack used was the $L_2$ attack described in Algorithm 1, with $P = 20, Q = 100, T = 5$. When adversarially attacking images smoothed with $q = 8192$ perturbations with this model, fewer perturbations are are used (512). In these experiments, the sample size is 144. Error bars represent the 95% confidence interval of the population mean. In Figure 4, instances where the adversarial attack failed were not counted at each point. For MNIST experiments in Section 4, the model architecture, attack method, and baseline

training method are the same as used for the adversarial training experiments, detailed in appendix E above. The adversarially training for classification uses the implementation provided in Rony et al. (2019), with default arguments.

# G  ADDITIONAL EXAMPLE IMAGES

See Figures 10 and 11.

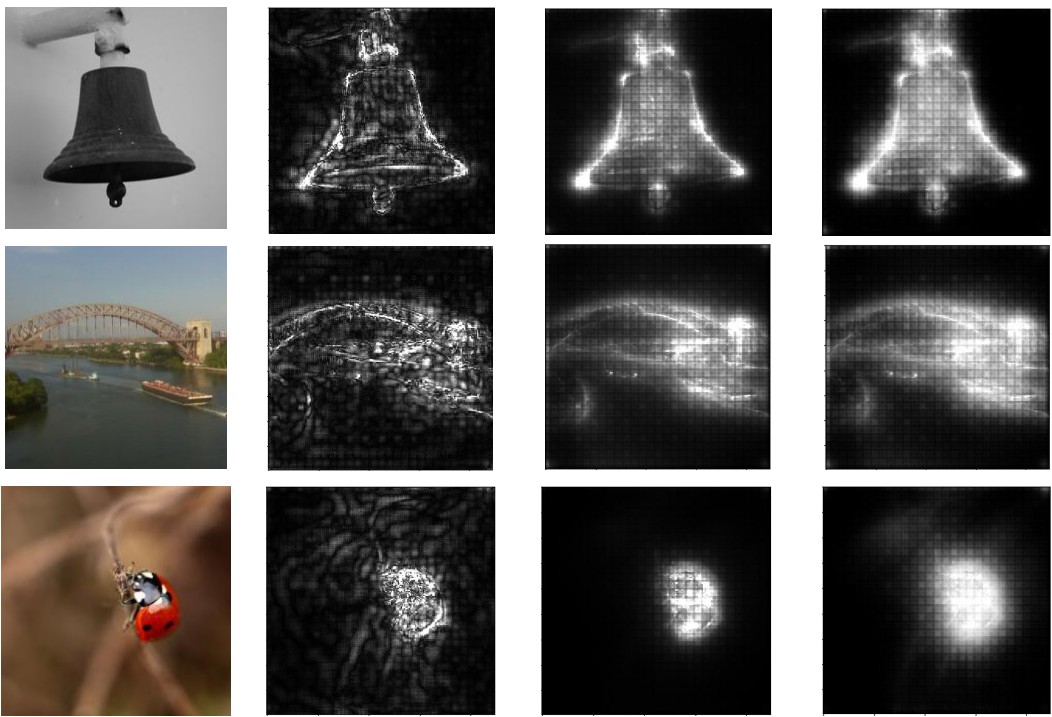

Figure 10: An illustration of different saliency maps on some images from ImageNet. The input image is shown in the first column (far left), with interpretations using SmoothGrad (second column from left), Quadratic SmoothGrad (third column), and Sparsified SmoothGrad ($\tau$ = .1, fourth column). $\sigma$ = .2, in units where pixel intensity ranges from 0 to 1.

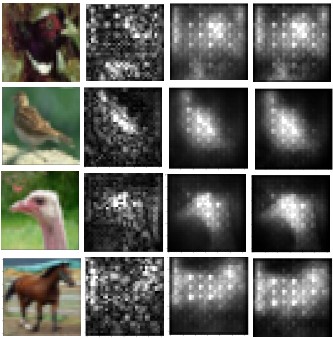

Figure 11: An illustration of different saliency maps on some images from CIFAR-10. The input image is shown in the first column (far left), with interpretations using SmoothGrad (second column from left), Quadratic SmoothGrad (third column), and Relaxed Sparsified SmoothGrad ($\tau$ = .1, $\gamma$ = .01, fourth column). $\sigma$ = .2, and the noise is scaled to the range of pixel intensities of the image.

## H BOUNDS FOR SCALED SMOOTHGRAD, QUADRATIC SMOOTHGRAD, AND RELAXED SPARSIFIED SMOOTHGRAD WITH $\gamma = 0$

In the text, we mention that we achieve vacuous bounds for Scaled SmoothGrad, Quadratic Smooth-Grad, and Relaxed Sparsified SmoothGrad with $\gamma = 0$. Here are these bounds (Figure 12):

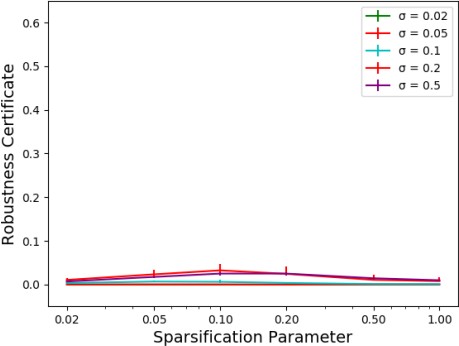

Figure 12: Bounds for Scaled SmoothGrad and and Relaxed Sparsified SmoothGrad with $\gamma = 0$. Note that Scaled SmoothGrad is equivalent to Relaxed Sparsified SmoothGrad with $\tau = 1$. Directly comparable to Figure 2. For the Quadratic case, the certificates were negligibly small: the largest robustness certificate achieved was 0.00023, at $\sigma = 0.2$.

## I ADDITIONAL IMAGES FROM ADVERSARIAL TRAINING EXPERIMENT (APPENDIX D)

See Figure 13.

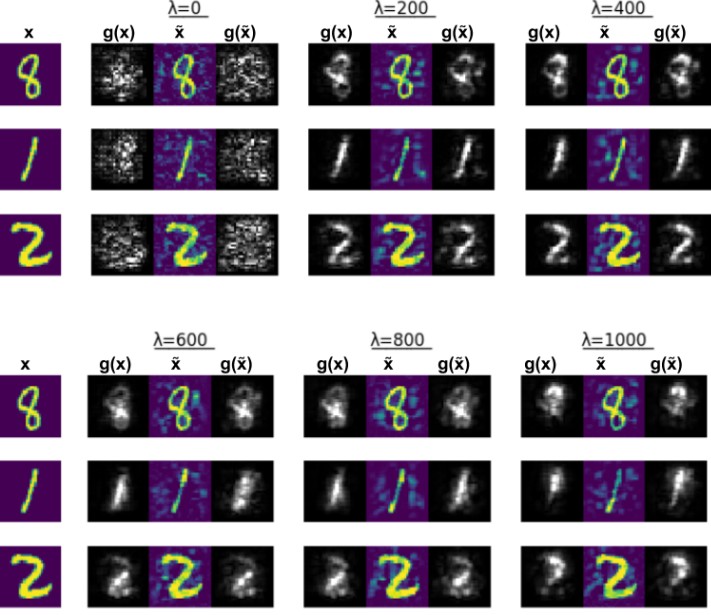

Figure 13: Additional figures from adversarial training on MNIST, for various $\lambda$. Note that Figure 7 shows for $\lambda = 200$.

## J    COMPARISON OF BOUNDS TO EMPIRICAL PERFORMANCE FOR RELAXED SPARSIFIED SMOOTHGRAD.

We present a detailed view of Figure 4, for small magnitude perturbations, with the robustness certificate shown. (Figure 14)

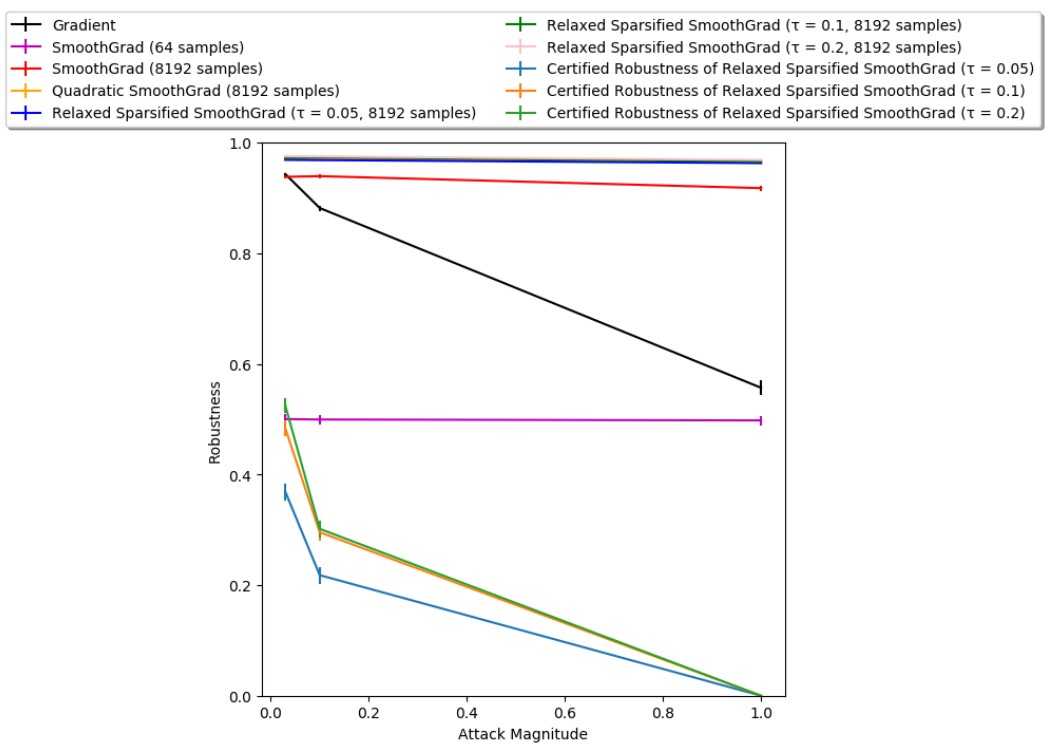

Figure 14: Empirical robustness of variants of SmoothGrad to adversarial attack, tested on CIFAR-10 with ResNet-18. Attack magnitude is in units of standard deviations of pixel intensity. Robustness is measured as $R(\mathbf{x}, \tilde{\mathbf{x}}, K)/K$, where $K = n/4$

## K    COMPARISON TO BOUNDS IN LECUYER ET AL. (2019)

Lecuyer et al. (2019) approaches the classification case for certified robustness by smoothing, by using bounds directly comparably to Theorem 1, but applying them to the class score elements, rather than the saliency map elements: bounds are certified by demonstrating that the top class score is certifiably larger than all other class scores. However, as noted by Cohen et al. (2019), these bounds are rather loose, and Cohen et al. (2019) gives significantly tighter bounds specifically for classification case, which we extend to apply to interpretation. In Figure 15, we compare our bounds for interpretation to a straightforward application of Lecuyer et al. (2019)'s results for class scores to saliency scores. Note that Lecuyer et al. (2019)'s results have a free parameter, for which we numerically maximize the bound.

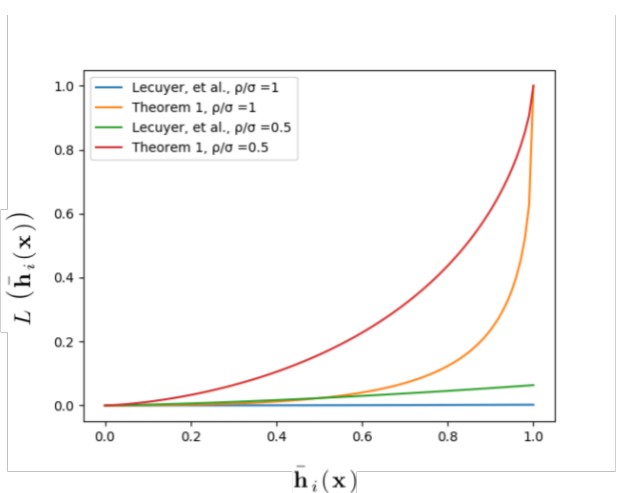

Figure 15: Comparison of Theorem 1 to results from Lecuyer et al. (2019)

