# OpenReview forum: "Certifiably Robust Interpretation in Deep Learning"
_ICLR.cc/2020/Conference — Reject_

### Official Review · AnonReviewer3 · 2019-10-22
**Official Blind Review #3**

**Rating:** 3

**Review:**

This paper introduces an extension of Cohen et al. (2019)’s result that allows one to derive robustness certificates for interpretation methods, as well as a bound on the top-K overlap of saliency methods. These results motivate the introduction of Sparsified SmoothGrad and a relaxation of this method that has differentiable elements. These introduced approaches adapt previous methods so the derived bounds are applicable. The proposed methods are shown to perform as well as Quadratic SmoothGrad (Smilkov et al. 2017) in CIFAR-10 experiments.

I’m not familiar with the field so it is hard for me to judge how novel the presented results are or whether the used baselines are the proper ones. That being said, the paper presents an interesting idea and it is relatively easy to read (I really appreciate the fact that for every theorem there is an interpretation, in words, for it). The only thing that sometimes makes the paper hard to read is when it starts to refer to too many constants without remind the reader what they are about. I have two complaints/questions about the relevance of the introduced bounds though. Right now, to me, it seems that the derived theoretical guarantees are not that relevant, hopefully the questions below will help clarify that.

In page 6, before introducing the “Sparsified SmoothGrad and its Relaxations”, it is said that q is set to 2^13 because otherwise the gap would be too large in images from ImageNet, for example, when comparing to traditional values of q. However, ImageNet is never revisited in the paper. I was expecting to see ImageNet results in the experimental section but they are not there (or maybe some correlation between the gap and performance -- robustness). More than that, the Quadratic SmoothGrad, which doesn’t have any theoretical guarantee, seems to perform as well as the proposed methods. So where is the gap/theoretical result relevant? What are the settings in which having a method with the derived theoretical guarantees shine? What are the limitations of Quadratic SmoothGrad? Right now, it seems to me that the “Sparsified SmoothGrad and its Relaxations” and its empirical analysis weaken the paper, because they take a big chunk of it when there is not enough evidence to claim them as an important contribution. Am I missing something? I gave this paper a relatively low score because I’m not certain about the relevance of its results, but if my questions are satisfactory answered, I’ll be happy to update my score.

------


>>> Update after rebuttal: I stand by my score after the rebuttal.

Unfortunately I'm not an expert in this area and I don't feel confident in having a very strong opinion about this paper. That being said, enough presentation issues were raised that make me uneasy about raising my score. I do agree with some of the concerns raised by other reviewers.


**Experience Assessment:**

I do not know much about this area.

**Review Assessment: Checking Correctness Of Derivations And Theory:**

I did not assess the derivations or theory.

**Review Assessment: Checking Correctness Of Experiments:**

I carefully checked the experiments.

**Review Assessment: Thoroughness In Paper Reading:**

I read the paper at least twice and used my best judgement in assessing the paper.

---

> ### Author Response · Authors · 2019-11-15
> **Author Response**
>
> We thank you for your feedback. We evaluate our robustness certificates on ImageNet samples in Figure 2. To address the concern about the gap between empirical and certified robustness, we show this gap on CIFAR samples in Appendix J. While the size of  perturbations with certified robustness is small compared to the empirical robustness on these samples, our main contribution is to demonstrate that a minor variation on the commonly-used SmoothGrad technique does in fact have a robustness guarantee: furthermore, this is the first robustness certificate for interpretation that can be evaluated at the ImageNet scale. This minor modification to SmoothGrad has little effect on the visual output (Figure 3).  Additionally, in testing the empirical attacks (Figure 4), we show that both quadratic SmoothGrad and our variant are empirically robust. Therefore, the variant (sparsified SmoothGrad) combines the visual quality and empirical robustness of Quadratic SmoothGrad with an additional theoretical guarantee of robustness.

---

### Official Review · AnonReviewer1 · 2019-10-23
**Official Blind Review #1**

**Rating:** 1

**Review:**

This paper proposes a way to testify how much a SmoothGrad saliency can vary from the true saliency attesting to the adversarial robustness but with the goal of interpretation.

At the premise of this work I do not think the paper motivates the value of such a robustness certificate. Using the gradient (with SmoothGrad), while providing a reasonable interpretation of the model, is just a linear approximation of the true explanation of the prediction. So saying we have the correct approximation is not so useful. I also am not sure we need such a method. For example imagine a doctor is looking at a saliency map and we are sure that it is correct first order approximation because of some method. What were the negative cases where this would fail? How would this method improve that? I believe right now just the basic gradient is sufficient to indicate the region of interest.


**Experience Assessment:**

I have published in this field for several years.

**Review Assessment: Checking Correctness Of Derivations And Theory:**

I assessed the sensibility of the derivations and theory.

**Review Assessment: Checking Correctness Of Experiments:**

I assessed the sensibility of the experiments.

**Review Assessment: Thoroughness In Paper Reading:**

I read the paper at least twice and used my best judgement in assessing the paper.

---

> ### Author Response · Authors · 2019-11-15
> **Author Response**
>
> We respectfully disagree. We believe that you may have misunderstood the main point of the paper. You mention that:
>
>  “I believe right now just the basic gradient is sufficient to indicate the region of interest.”
>
> The central issue here is that basic gradient methods may NOT in fact indicate the region of interest in an image. A small adversarial noise can keep the label as is but change the basic gradient result significantly. This is the problem that we are addressing in this paper.
>
> As you mention, gradient-based saliency maps represent only a local first order approximation to the true influence of each feature on the decision. This leads to two issues:
>
> * Low quality natural interpretations: as noted by Smilkov, et al. (2017) the gradient with respect to a particular pixel may “fluctuate sharply at small scales” and therefore be “less meaningful than a local average of gradient values.” This observation led to the development of SmoothGrad. To put this simply, a large gradient value over a (very) small range of input values of a feature represents in total a small influence on the class score by that feature. However, if the input image happens to be within this interval where the gradient is large, the feature will erroneously appear to be highly salient. In practice, this leads to simple gradient-based interpretations looking “noisy,” as apparently random pixels appear to be highly salient.
>
> * Adversarial attacks on interpretation: as demonstrated by Ghorbani, et al. (2019), one can adversarially craft examples where the basic gradient interpretation is in fact very different from the true region of interest. This is a direct consequence of the saliency map being a “first order approximation”: it is therefore possible to make this approximation adversarially bad, by crafting a small perturbation to the input.
>
> As detailed in the paper, saliency maps are used in a broad range of highly sensitive downstream applications, including in medical imaging and object localization. Because an adversarial attack has been proposed by Ghorbani et al. (2019) which can distort saliency maps, it is therefore a topic of interest to defend against this type of adversarial attack.

---

> > ### Comment · AnonReviewer1 · 2019-11-15
> > **still not a solid motivation**
> >
> > In your response the motivation you present now is that there is some adversary which will corrupt the image.
> >
> > I don't agree and I echo the "Rigor Police" comment here that there there is no reasonable adversary here for medical images. How can a criminal profit? Who is the criminal? What do they gain? It is important that the work have a solid motivation which clearly translates to move us forward so we don't spend time solving problems that people don't have.

---

### Official Review · AnonReviewer4 · 2019-10-27
**Official Blind Review #4**

**Rating:** 3

**Review:**

The work addresses an important problem of robustness of interpretation methods against adversarial perturbations. The problem is well motivated as several gradient-based interpretations are sensitive to small adversarial perturbations.

The authors present a framework to compute the robustness certificate (more precisely, a lower bound to the actual robustness) of any general saliency map over an input example. They further propose variants of SmoothGrad interpretation method which are claimed to be more robust.

The empirical validation of the underlying theory and use of the sparsified (and relaxed) SmoothGradient interpretation methods is unconvincing because of the following reasons:

1. In the demonstrated experiment, the proposed alternative to SmoothGrad involves setting the lowest 90% of the saliency values to zero, and the top 10% (for sparsified SmoothGrad) or top 1% (in the case of relaxed sparsified SmoothGrad) to one. The problem with clamping most of the lower values to zero and the remainder (or most of the remainder) higher values to one is that it defeats the purpose of having a saliency map in the first place, which exist to characterize the relative importance of the input features.

2. The paper claims that the proposed variant maintains the high visual quality of SmoothGrad, however, the claim is unsubstantiated. With the current setup, there is a clear trade-off between robustness and fidelity of interpretation, which the paper fails to acknowledge. In principle, one can always build extremely sparse or dense interpretation methods (close to all zeros or all ones), which would produce high robustness certificates but would be much less meaningful as they are not faithful to the underlying mechanism of prediction, and the characteristics of the input.

3. The authors present empirical evidence on just one set of sparsification parameters and K. It would be more conclusive to evaluate the robustness of the proposed variations with different values of sparsification parameters, and K.


**Experience Assessment:**

I have read many papers in this area.

**Review Assessment: Checking Correctness Of Derivations And Theory:**

I did not assess the derivations or theory.

**Review Assessment: Checking Correctness Of Experiments:**

I carefully checked the experiments.

**Review Assessment: Thoroughness In Paper Reading:**

I read the paper at least twice and used my best judgement in assessing the paper.

---

> ### Author Response · Authors · 2019-11-15
> **Author Response**
>
> We thank you for your constructive feedback. To address your comments:
>
> 1 and 2. Note that we sparsify the saliency maps before smoothing: in other words, the final smoothed saliency map will be non-sparse, because pixels which are less salient overall may still occur in the top 10% in a minority of random samples. Empirically, we find that this sparsification prior to averaging has little effect on the final smoothed interpretation: in particular, the results are visually very similar to the quadratic SmoothGrad proposed by (Smilkov et al. 2017). This was shown in Figure 3 on an ImageNet sample, as well as on additional CIFAR samples in Appendix G. To address this comment, we have added additional ImageNet samples both in the body of the paper (Figure 3) and in the appendix (Appendix G).
>
> 3: We have added empirical tests using additional values of the sparsification parameter to Figure 4.

---

### Public Comment · ~Rigor_Police1 · 2019-11-04
**motivation**

Recently, use of “medical applications” to motivate an idea has become a trend in machine learning. This paper is no different. The paper claims that gradient based saliency maps are used in several medical applications [paragraph 1] and these interpretation methods can be attacked. Really? Do you really believe that if neural network interpretation methods are used by doctors, attackers will gain access to these models that easily? And do you really believe that medical data can be tampered so easily to create adversarial attacks? Please think about plausibility of this attack and the legal implications this would have before making comments like these.

Let’s now say these interpretation methods used were somehow attacked. Do you really think doctors will blindly trust these systems and diagnose the patients? Doctors have had years of experience and these AI systems would merely be used to aid diagnosis. Whenever the system interprets something different, doctors would just ignore the suggestion. Before all this, there are millions of considerations on how AI should be used by doctors. Nobody knows this yet.

Another application where interpretation can be used is model debugging. But, this happens in development phase and adversarial attacks don’t make any sense here. So, I am not even convinced why addressing the problem of robust interpretation is important in the first place.

---

> ### Public Comment · ~Rigor_Police1 · 2019-11-04
> **method and empirical evaluation**
>
> The paper claims that Scaled smoothgrad and Quadratic smoothgrad give vacuous bounds. So, they develop a new method “Sparsified Smoothgrad”. And to qualitatively illlustrate this method, one example is provided on imagenet (the kind of datasets people really care about, nobody cares about MNIST) (Figure 1). With one example, how do we know if this newly developed method really performs well. Please see papers like Gradcam, where they show several examples to illustrate the interpretation methods.
>
> “We test Relaxed Sparsified SmoothGrad (γ = 0.01, τ = 0.1), rather than Sparsified SmoothGrad because our attack is gradient-based and Sparsified SmoothGrad has no defined gradients”. If methods don’t have explicit gradients, there are several attack strategies. Refer “Obfuscated gradients give a false sense of security” paper for more details.
>
> “We tested on ResNet-18 with CIFAR-10 with the attacker using a separately-trained, fully differentiable version of ResNet-18, with SoftPlus activations in place of ReLU”. Why can’t you use ReLU networks? Softplus networks are horrible and give really poor performance when trained. So obviously, the attacks are going to be sub-optimal when these models are used for creating attacks. And is there a reason why transfer attacks are used and not white-box attacks?
>
> Honestly, its hard to understand the experimental evaluation. Lot of notations. Empirical section is dense, and hard to parse. In Figure 2, what is the robustness certificate in y-axis? Is it the rank certificate? “The lines shown are for the 60th percentile guarantee, meaning that 60 percent of images had guarantees at least as tight as those shown”. What do you mean tight as those shown?
>
> Figure 2 is shown for the case K=0.2n, which is 20% of the entire image. Now, 20% is a big fraction of the image. Consider this case: An image has a small component contributing to a prediction, say 1% - this is not some example I made up for arguments’ sake. In several medical imaging and vision applications, this happens. Now, only 1% of the image is relevant in making prediction, and let us say gradient based saliency methods correct picked this top-1% overlap i.e., in the saliency map top 1% has high value, and others have very low value. By the bound you show in Figure 2, you guarantee that the prediction stays within 20%. For all you know, the method could highlight some noise, and push the 1% correct prediction to a low value (as the pixels other than 1% had low values in the original saliency map). In this case, the bound becomes useless. All I am saying is K is something that should not be picked before hand. And it is very important to analyze the certification rate as a function of K. 20% is still a very big number, and people really care about what happens for small K.
>
> The previous paragraph clearly states some issues with rank certificate. May be a better metric to look at is L_p norm between predicted and perturbed saliency maps?
>
> In my opinion, empirical evaluation is quite weak to access the importance of the approach. Attacks are created with Softplus network which are extremely weak in the first place. Whitebox setting is not considered. It’s hard to say the importance of provided bounds at high values of  K. Very few qualitative results are presented at imagenet scale. Effect of certification as a function of K is not analyzed. MNIST and CIFAR-10 are simple classification tasks with small images, so the results obtained here are not reflective of what happens as the size of images increase to Imagenet or COCO scale. Rank certification by itself can lead to issues, so it’s not clear if this is even the right form of certification to look at.

---

> > ### Public Comment · ~Simon_Kornblith1 · 2019-11-05
> > **Softplus networks are not horrible**
> >
> > Dear Mr. Police,
> >
> > You say "Softplus networks are horrible." This statement is extremely unfair to softplus networks. The Swish paper [1] provides comprehensive results for 9 networks with different activation functions. Softplus often outperforms ReLU.
> >
> > I have not read this paper and this comment is not an endorsement of anything besides softplus networks.
> >
> > [1] Ramachandran, P., Zoph, B., & Le, Q. V. (2017). Searching for activation functions. https://arxiv.org/abs/1710.05941

---

> > ### Author Response · Authors · 2019-11-15
> > **Author Response**
> >
> >  We thank you for your comment.
> >
> > Qualitative Evaluation on ImageNet: We have added additional qualitative comparisons on ImageNet, in Figure 3 and Appendix G.
> >
> > Gradients, SoftPlus and Transfer vs Whitebox Attacks: In order to use first-order methods to adversarially attack gradient-based interpretations, a network must have defined second derivatives with respect to the input image (because the saliency map itself consists of the first derivatives of the output with respect to the input image).  ReLU networks thus cannot be attacked in this way. Therefore, we use a proxy network with SoftPlus activations to determine the direction of the attack.
> >
> > Figure 2 Y Axis: This is the 60th percentile of the robustness certificate: 60 percent of images have robustness certificates at least this large.
> >
> > Rank-based Certificates: it is clear that an $L_p$ norm based metric would be inappropriate for the purpose of certifying similarity between saliency maps: in most works using gradient-based saliency maps (e.g., Sundararajan et al.  (2017)), the top values are clipped for visualization purposes, so that the rest of the interpretation can be scaled to a reasonable color range without being dominated by a few large outlier pixels. This suggests that an $L_p$ norm approach may be meaningless, because an $L_p$ norm could be dominated by the behavior of outlier values. The fact that this clipping is accepted practice also indicates that it is the relative rank of the importance of features of an image, rather than the absolute ratios between salience measures, that is important for interpretation.

---

> ### Author Response · Authors · 2019-11-15
> **Author Response**
>
> We thank you for your comment. Your concerns here are equally applicable to the study of adversarial robustness in the classification case: in both instances, there is a desire to protect against adversarial attacks which may affect how a machine learning system makes decisions. The suggestion that users would “ignore the decision” if a system returns an incorrect result assumes that the system is entirely redundant: that its output has no effect on the users’ behavior. This is true in the classification case as well: if we a priori assume that the users know the correct classification before looking at the output, adversarial examples cannot possibly cause any harm. In addition to the medical examples laid out in the paper, gradient-based methods are also used for automated image segmentation and object localization: Subramanya, et al. (https://arxiv.org/abs/1812.02843) recently introduced an adversarial attack against GradCAM, a variation of gradient-based saliency maps which is tailored for object localization specifically.
> In the classification case, the wide literature on adversarial robustness published in recent years indicates that  the community considers adversarial attacks to be an issue worthy of concern: attacks against interpretation are just as plausible from a security standpoint as (non-physical) attacks against classification.

---

### Decision · Program_Chairs · 2019-12-19

**Decision:**

Reject

**Comment:**

This paper discusses new methods to perform adversarial attacks on salience maps.

In its current form, this paper in its current form has unfortunately has not convinced several of the reviewers/commenters of the motivation behind proposing such a method. I tend to share the same opinion. I would encourage the authors to re-think the motivation of the work, and if there are indeed solid use cases to express them explicitly in the next version of the paper.

---

> ### Author Response · Authors · 2019-12-21
> **Author Response**
>
> This review, unfortunately, mischaracterizes the main contribution of our paper. We propose a provable *defense* against adversarial attacks on saliency maps: such attacks were already previously proposed by other authors (Ghorbani et al. 2019). The existence of these attacks provides the motivation for provable defenses, e.g. our work.